# ANCHORED SUPERVISED FINE-TUNING

**He Zhu**[1,2]*, **Junyou Su**[2]*, **Peng Lai**[1]*, **Ren Ma**[3], **Wenjia Zhang**[2], **Linyi Yang**[1], **Guanhua Chen**[1]†
[1]Southern University of Science and Technology, [2]Peking University
[3]Shanghai Artificial Intelligence Laboratory
{zhuhe, jysu25}@stu.pku.edu.cn, chengh3@sustech.edu.cn

## ABSTRACT

Post-training of large language models involves a fundamental trade-off between supervised fine-tuning (SFT), which efficiently mimics demonstrations but tends to memorize, and reinforcement learning (RL), which achieves better generalization at higher computational cost. Dynamic Fine-Tuning (DFT) recently emerged as a promising middle ground, reweighting SFT objectives with token probabilities and achieving improvements in certain reasoning domains, though it exhibits instability in other tasks. We provide an analysis of DFT through the **reward-weighted regression (RWR)** framework, revealing that it corresponds to a specific auxiliary distribution choice that yields provably tighter RL bounds than standard SFT. However, our analysis also uncovers a critical limitation: this construction lacks distributional anchoring, leading to progressive drift that undermines training stability. To address this, we propose **Anchored Supervised Fine-Tuning (ASFT)**, which augments DFT's reweighting with **lightweight KL regularization** to preserve tightness while ensuring stability. Empirically, ASFT consistently outperforms both SFT and DFT across mathematical reasoning, medical knowledge grounding, and code generation, achieving substantial improvements with minimal computational overhead. Our RWR framework provides a systematic lens for understanding post-training methods and demonstrates that principled theoretical analysis leads to both stronger guarantees and practical gains. [1]

## 1 INTRODUCTION

Large language models (LLMs) have become a central substrate for modern AI systems, powering instruction following, tool use, and multi-step reasoning at scale (Brown et al., 2020; Touvron et al., 2023; Achiam et al., 2023; Guo et al., 2025). Post-training is crucial to adapt pretrained models to tasks and human preferences. This process typically involves two primary paradigms: supervised fine-tuning (SFT), which is an off-policy method that imitates expert demonstrations collected from a fixed dataset, and reinforcement learning (RL), which is an on-policy approach that optimizes outcome-based rewards by directly interacting with the model's own outputs (Ouyang et al., 2022; Rafailov et al., 2023; Shao et al., 2024). While SFT is data- and compute-efficient, excelling at rapid acquisition of desired behaviors, it tends to memorize surface patterns rather than learn robust, generalizable strategies (Chu et al., 2025; Zhang et al., 2021; Feldman, 2020). In contrast, RL leverages outcome-driven updates and exploration to discover more transferable behaviors, but is substantially more expensive and unstable in practice (Schulman et al., 2017; Ziegler et al., 2019; Stiennon et al., 2020). This fundamental trade-off motivates methods that retain SFT's efficiency while inheriting RL's generalization benefits (Bai et al., 2022; Lee et al., 2023; Yuan et al., 2024).

A growing body of work re-examines SFT through an RL lens, arguing that the *implicit reward* induced by maximum likelihood is pathological and that principled reweighting or trust regions are needed. Among these approaches, Dynamic Fine-Tuning (DFT) (Wu et al., 2025a) has gained significant attention by identifying a pathological reward structure in standard SFT that leads to unbounded variance when model probabilities approach zero. DFT addresses this through probability-based reweighting, achieving remarkable empirical improvements in mathematical reasoning tasks. However, our preliminary experiments reveal that DFT's effectiveness is domain-specific; it excels

---

*Equal Contribution. †Corresponding Author.
[1]The code is available at https://github.com/zhuchichi56/ASFT.

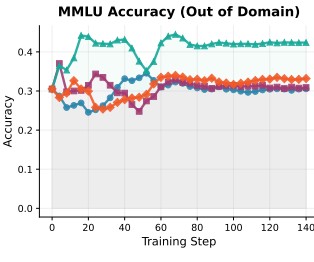 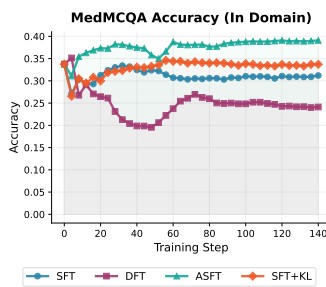 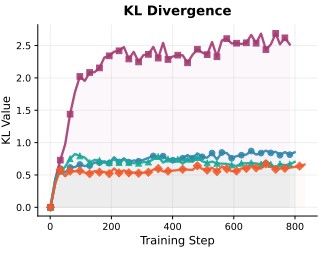

Figure 1: Training dynamics comparison across fine-tuning methods on medical knowledge tasks. **Left**: MMLU accuracy (out-of-domain evaluation); **Center**: MedMCQA accuracy (in-domain evaluation); **Right**: KL divergence from base model. DFT exhibits severe distributional drift (high KL divergence) while ASFT maintains stability through KL anchoring and achieves superior performance on both tasks.

in reasoning-intensive domains yet exhibits instability in knowledge-intensive tasks and lacks theoretical grounding for its design choices.

To address these problems, we provide a principled theoretical analysis of DFT within a reward-weighted regression framework inspired by reward-weighted regression and importance sampling theory (Rubinstein & Kroese, 2016). From the theoretical perspective, we reveal that DFT corresponds to a specific auxiliary distribution construction that yields a provably tighter lower bound on the RL objective compared to standard SFT. This causes a critical limitation: the absence of distributional anchoring mechanisms leads to progressive drift away from the reference distribution. The distributional shift makes the bound increasingly loose and increases importance weight variance, explaining the training instabilities of DFT.

To address these fundamental limitations, we propose **Anchored Supervised Fine-Tuning (ASFT)**, a lightweight extension of DFT that incorporates a KL divergence regularization term to prevent distributional drift while preserving the tightness benefits of adaptive reweighting. As demonstrated in Figure 1, ASFT maintains stable KL divergence while achieving superior performance across both in-domain and out-of-domain evaluations.

Empirically, ASFT consistently outperforms both standard SFT and DFT across mathematical reasoning, medical knowledge, and code generation tasks. On mathematical reasoning benchmarks with 100k training samples, ASFT achieves an average improvement of +4.85 points (18.6%) over DFT and +17.89 points (142%) over the base model. In medical knowledge tasks with 10k samples, ASFT delivers +8.28 points (24.8%) improvement over SFT and +10.65 points (33.9%) over the base model, requiring only 3% of the training cost of full RL approaches.

**Our contributions** are threefold: **(1)** We provide a rigorous theoretical explanation for DFT's domain-specific effectiveness and inherent instabilities, grounding its heuristic design within the formal reward-weighted regression framework and proving that it achieves a strictly tighter bound than SFT while suffering from uncontrolled variance growth. **(2)** We propose ASFT, a simple yet principled method that resolves DFT's stability issues through lightweight KL anchoring while maintaining its tightness advantages, requiring minimal computational overhead compared to full RL approaches. **(3)** We demonstrate that ASFT delivers superior performance across both reasoning-intensive and knowledge-intensive domains, achieving better generalization than SFT, greater stability and broader applicability than DFT, and RL-comparable performance with SFT-level computational efficiency.

## 2 RELATED WORK

**Supervised fine-tuning and reinforcement learning.** Supervised fine-tuning (SFT) and reinforcement learning (RL) are the two dominant paradigms for post-training large language models. SFT can be viewed as optimizing a stable but *loose lower bound* on the RL objective, which ensures efficiency and robustness but often leads to memorization and limited generalization (Wei et al.,

2022; Chung et al., 2024; Zhang et al., 2021; Chu et al., 2025). In contrast, RL directly optimizes outcome-based rewards, achieving tighter alignment and stronger generalization but at the cost of instability, high variance, and heavy computation (Ouyang et al., 2022; Schulman et al., 2015; 2017). Recent work seeks to bridge this trade-off by either tightening the SFT bound via reweighting and importance weighting (Wu et al., 2025a; Qin & Springenberg, 2025) or stabilizing RL through trust-region and hybrid methods (Sheng et al., 2025; Zhu et al., 2025).

**Importance weighting and policy optimization.** The connection between supervised learning and reinforcement learning through importance weighting has deep theoretical roots (Kahn & Marshall, 1953; Dayan & Hinton, 1997). In the context of policy optimization, importance sampling enables off-policy learning by reweighting samples from a behavior policy to estimate gradients for a target policy (Metelli et al., 2018; Jiang & Li, 2016), though the resulting weights can suffer from high variance when distributions are misaligned (Andradóttir et al., 1995). To address this, trust-region methods (Schulman et al., 2015) and proximal policy optimization (Schulman et al., 2017) constrain policy updates to remain close to a reference policy. Building on these ideas, recent advances in language model fine-tuning have explored importance-weighted supervised objectives (Qin & Springenberg, 2025), proximal supervised fine-tuning (Zhu et al., 2025), and probability-based reweighting in dynamic fine-tuning (Wu et al., 2025a), as well as unified frameworks that combine SFT and RL principles (Lv et al., 2025; Wu et al., 2025b). Our work follows this line of research by anchoring the auxiliary distribution itself to the base model, thereby extending importance-weighted approaches with a mechanism that stabilizes training while retaining their theoretical tightness.

## 3 PRELIMINARIES

In this section, we establish the theoretical foundation connecting supervised fine-tuning and reinforcement learning through the reward-weighted regression framework, setting the stage for our analysis of existing methods and our proposed ASFT approach.

### 3.1 PROBLEM FORMULATION AND BASIC FRAMEWORKS

We consider language modeling where trajectories $\tau = (x, y)$ consist of input prompts $x$ and generated responses $y$. A parametric policy $\pi_\theta(\tau) = \pi_\theta(y \mid x) = \prod_{t=1}^{|y|} \pi_\theta(y_t \mid y_{<t}, x)$ assigns probability via autoregressive decomposition. The **reinforcement learning objective** maximizes expected reward $J(\theta) = \mathbb{E}_{\tau \sim \pi_\theta}[R(\tau)]$ where $R(\tau) : \mathcal{X} \times \mathcal{Y} \to [0, 1]$ evaluates trajectory quality. In contrast, **supervised fine-tuning** performs behavior cloning on expert demonstrations $\mathcal{D} = \{(x, y^*)\}$ sampled from reference distribution $\pi_{\text{ref}}(\tau)$ by minimizing $\mathcal{L}_{\text{SFT}}(\theta) = -\mathbb{E}_{(x,y^*) \sim \mathcal{D}}[\log \pi_\theta(y^* \mid x)]$.

### 3.2 THE REWARD-WEIGHTED REGRESSION FRAMEWORK

Building on prior work in reward-weighted regression and importance sampling (Peters & Schaal, 2007; Rubinstein & Kroese, 2016; Qin & Springenberg, 2025), we adopt the **reward-weighted regression (RWR) framework** for language model fine-tuning. This framework provides a principled connection between SFT and RL objectives by leveraging importance sampling and auxiliary distributions to construct tighter bounds on the RL objective.

Under sparse rewards where $R(\tau) = \mathbb{I}[y = y^*]$ and the assumption that $\text{supp}(\pi_\theta) \subseteq \text{supp}(\pi_{\text{ref}})$, we can establish the following fundamental result:

**Proposition 1** (SFT as RL Lower Bound)**.** *The RL objective satisfies:*

$$J(\theta) \geq c_{\text{ref}} \cdot \mathbb{E}_{\tau \in D^+}[\log \pi_\theta(\tau)] \tag{1}$$

*where $D^+ = \{(x, y^*) \mid R(x, y^*) = 1\}$ and $c_{\text{ref}} = \mathbb{P}_{\pi_{\text{ref}}}(\tau \in D^+)$.*

This reveals that SFT optimization implicitly maximizes a lower bound on the RL objective. However, this bound becomes increasingly loose as $\pi_\theta$ diverges from $\pi_{\text{ref}}$ during training.

Within the RWR framework, we can generalize to tighter bounds through auxiliary distributions. For any distribution $q(\tau)$ with appropriate support:

$$J(\theta) \geq c_{\text{ref}} \cdot \mathbb{E}_{\tau \in D^+} \left[ \frac{q(\tau)}{\pi_{\text{ref}}(\tau)} \log \pi_\theta(\tau) \right] \tag{2}$$

The choice of auxiliary distribution $q$ determines both the tightness of the bound and the stability of the resulting optimization procedure. This sets up the fundamental trade-off between validity and tightness that we address in this work.

### 3.3 DYNAMIC FINE-TUNING: AN EXISTING APPROACH

Dynamic Fine-Tuning (DFT) (Wu et al., 2025a) addresses SFT's limitations by identifying a pathological reward structure in standard SFT. When viewed as a policy gradient method, SFT's implicit reward $r_{\text{SFT}}(y \mid x) = \frac{\mathbb{I}[y=y^*]}{\pi_\theta(y|x)}$ exhibits inverse-probability weighting that causes unbounded variance when $\pi_\theta(y^* \mid x)$ approaches zero.

DFT addresses this through probability-based reweighting:

$$\mathcal{L}_{\text{DFT}}(\theta) = -\mathbb{E}_{(x,y^*)\sim\mathcal{D}}[\text{sg}[\pi_\theta(y^* \mid x)] \log \pi_\theta(y^* \mid x)] \tag{3}$$

where $\text{sg}[\cdot]$ denotes the stop-gradient operator.

While DFT achieves empirical improvements, its theoretical properties within the RWR framework remained unclear—a gap we address in our analysis.

## 4 METHOD

### 4.1 THEORETICAL ANALYSIS OF DFT WITHIN THE RWR FRAMEWORK

We found that DFT can be precisely characterized within the RWR framework through a specific auxiliary distribution construction. This analysis reveals both DFT's strengths and fundamental limitations.

**Key Finding 1: DFT corresponds to a specific auxiliary distribution choice.** We discovered that the DFT objective is mathematically equivalent to choosing the auxiliary distribution:

$$q(\tau) = \frac{\pi_{\text{ref}}(\tau \mid D^+) \, \text{sg}[p_\theta(\tau)]}{\mathbb{E}_{\tau \sim \pi_{\text{ref}}(\cdot|D^+)}[\text{sg}[p_\theta(\tau)]]} \tag{4}$$

This construction directly recovers the DFT sequence-level objective:

$$\mathcal{L}_{\text{DFT}}(\theta) = -\mathbb{E}_{\tau \in D^+}[\text{sg}(p_\theta(\tau)) \log p_\theta(\tau)] \tag{5}$$

**Key Finding 2: DFT achieves provably tighter bounds than SFT.** We proved that this auxiliary distribution yields a strictly tighter lower bound on the RL objective compared to standard SFT whenever the policy assigns non-uniform probabilities to demonstrations (detailed proof in Appendix D.4).

**Theorem 1** (Strict Tightness). *The DFT auxiliary distribution yields a strictly tighter lower bound than standard SFT whenever* $\text{Var}(p_\theta(\tau)) > 0$ *on* $D^+$.

This theoretical result explains DFT's superior empirical performance in domains where the policy distribution exhibits sufficient variance across training examples.

**Key Finding 3: DFT suffers from distributional drift.** However, our analysis also revealed a critical limitation: the policy distribution progressively diverges from the reference distribution during training. As optimization proceeds, $q$ becomes increasingly concentrated on trajectories with high $p_\theta(\tau)$, creating a feedback loop where the model focuses on a diminishing subset of training data. This distributional drift makes the bound increasingly loose and the importance weights increasingly high-variance, reducing effective sample size and destabilizing optimization.

We formalized this instability by noting that the fundamental inequality $u \geq 1 + \log u$ (used to derive the RL lower bound) achieves equality if and only if $u = 1$. In DFT's case, $u = \frac{\pi_\theta(\tau)}{q_\theta(\tau)}$, so the bound is tight only when $\pi_\theta(\tau) = q_\theta(\tau)$, i.e., when $p_\theta(\tau)$ is constant on $D^+$. However, as training progresses, $p_\theta(\tau)$ becomes increasingly non-uniform, making the inequality strictly loose. This leads to deteriorating bound quality, reduced effective sample size, and training instability.

## 4.2 ANCHORED SUPERVISED FINE-TUNING (ASFT)

To address DFT's distributional drift while preserving its tightness advantages, we propose **Anchored Supervised Fine-Tuning (ASFT)**. Our method adds a lightweight KL regularization term that constrains the policy within a trust region of a reference checkpoint:

$$\mathcal{L}_{\mathrm{ASFT}}(\theta) = \mathcal{L}_{\mathrm{DFT}}(\theta) + \lambda \mathbb{E}_s[D_{\mathrm{KL}}(\pi_{\mathrm{base}}(\cdot \mid s) \| \pi_\theta(\cdot \mid s))] \tag{6}$$

where $\pi_{\mathrm{base}}$ is a fixed reference policy (typically the pretrained model) and $\lambda > 0$ controls anchoring strength.

**Theoretical Guarantees.** This design preserves DFT's tightness benefits since the KL term does not alter the lower-bound structure, while providing explicit variance control that prevents the exponential growth that destabilizes pure DFT training. The anchoring mechanism creates a trust region around the reference policy, allowing controlled exploration of tighter bounds without sacrificing distributional stability.

**Practical Implementation.** Following standard practice in language model training (Ouyang et al., 2022; Shao et al., 2024), we implement ASFT at the token level by distributing sequence-level weights across tokens using normalized per-position allocation, ensuring mathematical equivalence to our theoretical framework while enabling efficient computation. Our method requires minimal computational overhead compared to standard SFT - adding only a simple KL penalty - yet delivers RL-comparable generalization performance along with SFT-level efficiency.

## 5 EXPERIMENTS

### 5.1 SETUP

**Models.** We conduct fine-tuning experiments using LLaMA-2-7B(Touvron et al., 2023) and Qwen2.5-7B(Qwen et al., 2025), two widely adopted models in the field. We select LLaMA-2-7B specifically to avoid potential contamination from prior supervised knowledge. Qwen2.5-7B, on the other hand, is a state-of-the-art model that is broadly used in current research. For knowledge-intensive (medical) tasks, we utilize both LLaMA-2-7B and Qwen2.5-7B to evaluate knowledge fine-tuning. For mathematical reasoning tasks, we focus exclusively on Qwen2.5-7B due to its superior reasoning capabilities, whereas LLaMA-2-7B is less competitive in mathematical reasoning. This setup enables a systematic study of knowledge and reasoning learning across both fact-based and reasoning-intensive domains.

**Datasets.** We evaluate ASFT on two domains: (i) **Mathematical reasoning**, using 10k/30k/100k samples from NuminaMath CoT (LI et al., 2024) for training, and testing on Math500 (Hendrycks et al., 2021), Minerva Math (Lewkowycz et al., 2022), OlympiadBench (AI Mathematical Olympiad, 2024), AIME 2024 (American Institute of Mathematics, 2024), and AMC 2023 (Mathematical Association of America, 2023); (ii) **Medical knowledge**, using 10k/30k/100k MedMCQA (Pal et al., 2022) samples for training, and testing on MMLU-medical (Hendrycks et al., 2020), MedQA (Jin et al., 2021), and the MedMCQA test set.

**Training and Evaluation Settings.** All methods are implemented using the AdamW optimizer with a constant learning rate. For **mathematical reasoning**, SFT and DFT use model_max_length 2048, global_batch_size 256, learning_rate $5 \times 10^{-5}$, and are trained for 1 epoch. ASFT follows the same configuration with coefficient $\lambda = 0.05$. For **medical knowledge**, we set model_max_length 512, global_batch_size 64, learning_rate $2 \times 10^{-5}$, and train for 3 epochs, with ASFT again using $\lambda = 0.05$. At the evaluation stage, for math, we use the default chat template and Chain-of-Thought

| Methods | Medical Benchmarks | | | | Math Benchmarks | | | | | |
|---|---|---|---|---|---|---|---|---|---|---|
| | MedQA | MMLU | MedMCQA | Avg. | AIME24 | Math500 | Minerva | Olympiad | ACM23 | Avg. |
| Base | 29.85 | 30.52 | 33.76 | 31.38 | 1.65 | 28.79 | 9.26 | 7.69 | 15.65 | 12.61 |
| *Dataset Scale: 10k* | | | | | | | | | | |
| SFT | 33.31 ↑3.46 | 33.52 ↑3.00 | 33.28 ↓0.48 | 33.37 ↑1.99 | 1.24 ↓0.41 | 41.84 ↑13.05 | 11.30 ↑2.04 | 12.26 ↑4.57 | 17.03 ↑1.38 | 16.73 ↑4.12 |
| SFT w/ KL | 29.22 ↓0.63 | 30.63 ↑0.11 | 33.01 ↓0.75 | 30.95 ↓0.43 | 0.41 ↓1.24 | 42.21 ↑13.42 | 12.05 ↑2.79 | 12.08 ↑4.39 | 17.19 ↑1.54 | 16.79 ↑4.18 |
| DFT | 29.69 ↓0.16 | 26.69 ↓3.83 | 31.20 ↓2.56 | 29.19 ↓2.19 | 4.18 ↑2.53 | 59.51 ↑30.72 | 17.10 ↑7.84 | 24.95 ↑17.26 | 31.09 ↑15.44 | 27.77 ↑15.16 |
| **ASFT** | **39.28** ↑9.43 | **46.37** ↑15.85 | **40.45** ↑6.69 | **42.03** ↑10.65 | 3.33 ↑1.68 | 59.60 ↑30.81 | **19.91** ↑10.65 | 24.50 ↑16.81 | **36.41** ↑20.76 | **28.75** ↑16.14 |
| *Dataset Scale: 30k* | | | | | | | | | | |
| SFT | 33.54 ↑3.69 | 38.48 ↑7.96 | 36.03 ↑2.27 | 36.02 ↑4.64 | 2.71 ↑1.06 | 44.74 ↑15.95 | 13.21 ↑3.95 | 13.44 ↑5.75 | 21.56 ↑5.91 | 19.93 ↑7.32 |
| SFT w/ KL | 30.56 ↑0.71 | 29.86 ↓0.66 | 33.56 ↓0.20 | 31.33 ↓0.05 | 2.70 ↑1.05 | 44.91 ↑16.12 | 13.03 ↑3.77 | 13.48 ↑5.79 | 18.90 ↑3.25 | 18.60 ↑5.99 |
| DFT | 31.26 ↑1.41 | 33.08 ↑2.56 | 35.09 ↑1.33 | 33.14 ↑1.76 | 3.34 ↑1.69 | 57.93 ↑29.14 | 23.28 ↑14.02 | 25.31 ↑17.62 | 28.44 ↑12.79 | 27.66 ↑15.05 |
| **ASFT** | **42.03** ↑12.18 | **44.94** ↑14.42 | **39.06** ↑5.30 | **42.01** ↑10.63 | **5.81** ↑4.16 | 57.03 ↑28.24 | 20.61 ↑11.35 | 24.44 ↑16.75 | **30.00** ↑14.35 | 27.18 ↑14.57 |
| *Dataset Scale: 100k* | | | | | | | | | | |
| SFT | 33.46 ↑3.61 | 38.01 ↑7.49 | 35.67 ↑1.91 | 35.71 ↑4.33 | 0.83 ↓0.82 | 47.30 ↑18.51 | 13.46 ↑4.20 | 14.16 ↑6.47 | 20.00 ↑4.35 | 19.15 ↑6.54 |
| SFT w/ KL | 30.09 ↑0.24 | 31.62 ↑1.10 | 33.85 ↑0.09 | 31.85 ↑0.47 | 1.44 ↓0.21 | 46.81 ↑17.02 | 14.13 ↑4.87 | 13.74 ↑6.05 | 20.00 ↑4.35 | 19.22 ↑6.61 |
| DFT | 36.61 ↑6.76 | 41.26 ↑10.74 | 36.31 ↑2.55 | 38.06 ↑6.68 | 6.26 ↑4.61 | 56.88 ↑28.09 | 21.18 ↑11.92 | 22.68 ↑14.99 | 27.19 ↑11.54 | 26.04 ↑13.43 |
| **ASFT** | **40.61** ↑10.76 | **42.02** ↑11.50 | **37.32** ↑3.56 | **39.98** ↑8.60 | **6.66** ↑5.01 | **59.99** ↑31.20 | **23.55** ↑14.29 | **25.57** ↑17.88 | **36.72** ↑21.07 | **30.50** ↑17.89 |

Table 1: Performance comparison of fine-tuning methods on medical benchmarks (left, base: LLaMA-2-7B) and math benchmarks (right, base: Qwen2.5-7B) under different dataset scales. **Bold** numbers indicate the best performance in each group, and rows with blue background highlight our ASFT approach. Arrows with ↑ and ↓ indicate performance improvements and degradations relative to the base model.

(CoT) prompting, report average accuracy over 16 decoding runs (temperature 1.0, max length 4096). For medical, we use standard prompt templates and multiple-choice accuracy. Baselines include SFT, SFT w/ KL, and DFT (Wu et al., 2025b).

## 5.2 MAIN RESULTS

Our experimental results demonstrate that ASFT consistently delivers superior performance across both knowledge-intensive and reasoning-intensive domains while maintaining training stability. As shown in Table 1, across both medical knowledge and mathematical reasoning tasks, ASFT consistently delivers strong and stable improvements over all baselines. In knowledge-intensive domains, ASFT not only avoids the severe performance degradation observed with DFT (which drops by an average of -2.19 points at 10k samples), but also achieves substantial gains—outperforming the base model by +10.65 points (a 33.9% relative improvement) at 10k scale, and maintaining robust advantages as the dataset size increases (10k: +10.65, 30k: +10.63, 100k: +8.60). This stability across scales highlights ASFT 's scalability and addresses the distributional drift issues that limit DFT in such settings. For mathematical reasoning, both DFT and ASFT surpass standard SFT, but ASFT maintains a consistent edge and greater training stability. With 100k samples, ASFT improves over the base by +17.89 points (versus DFT's +13.43), and the advantage is even more pronounced on challenging benchmarks like AMC23 (36.72% vs. 27.19% for DFT), reflecting superior generalization. Overall, ASFT 's improvements are not only larger but also more consistent across diverse benchmarks, while DFT's gains are more variable—especially on tasks requiring broad mathematical reasoning—underscoring the robustness and effectiveness of ASFT.

## 5.3 ABLATION STUDY

We conduct two ablations: (1) comparing forward vs. reverse KL regularization, and (2) analyzing the impact of learning rate and batch size. These studies clarify the effect of KL direction and the robustness of ASFT to key hyperparameters.

**Forward vs. Reverse KL.** Here we follow the standard notation: let $Q$ denote the policy model (i.e., the fine-tuned model with parameters $\theta$), and $P$ denote the reference model (the pretrained base model). In our approach, we use the **forward** KL divergence, $D_{\text{KL}}(P \parallel Q)$, to regularize the policy model. For comparison, we also experiment with the **reverse** KL divergence, $D_{\text{KL}}(Q \parallel P)$. As shown in Figure 2, forward KL leads to consistent improvements. Theoretically, forward KL encourages **mode-covering** behavior, preventing the model from collapsing and ensuring it maintains the broad distribution of the base model. In contrast, reverse KL is **mode-seeking**, which can cause

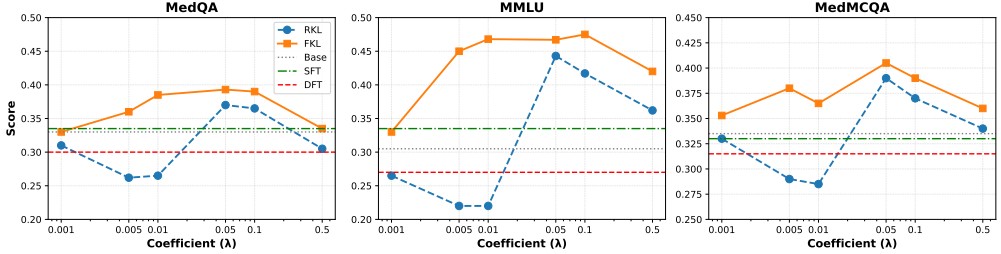

Figure 2: Comparison of forward KL (FKL) and reverse KL (RKL) regularization effects across different coefficient values ($\lambda$) on MedQA, MMLU, and MedMCQA benchmarks. Performance is measured in accuracy, with horizontal dashed lines indicating baseline performance of Base, SFT, and DFT models.

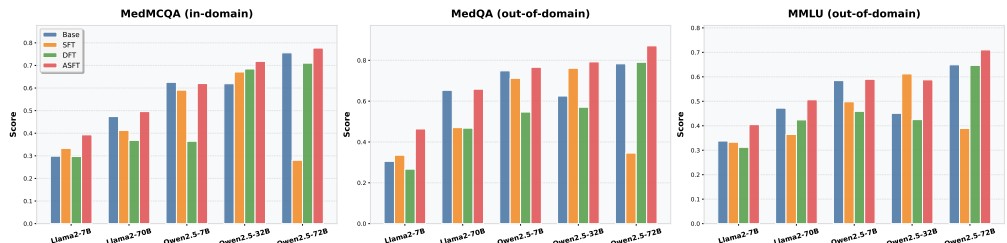

Figure 3: Comparison of model performance across three benchmarks (MedQA, MMLU, MedM-CQA) for five models (LLaMA-2-7B, LLaMA-2-70B, Qwen2.5-7B, Qwen2.5-32B, Qwen2.5-72B) using four fine-tuning strategies (Base, SFT, DFT, ASFT). Each subplot shows the scores for a specific benchmark, highlighting the relative effectiveness of different fine-tuning methods across models.

the model to focus excessively on a few high-probability sequences. We further sweep the regularization coefficient $\lambda$ and find that the optimal range is around $\lambda = 0.1$; both excessively small and large values lead to under-anchoring or over-regularization, respectively.

**Training Hyper-Parameters Ablation.** We conduct an ablation study to evaluate the robustness of ASFT with respect to learning rate and batch size, using LLaMA-2-7B and 10k samples from MedCAQA. For learning rate, we examine six values ranging from 5e-6 to 2e-4. ASFT consistently outperforms both SFT and DFT across all configurations, with the best performance observed at intermediate rates (1e-5 and 2e-4). Extreme low (5e-6) or high (2e-4) rates lead to minor drops, highlighting that moderate learning rates are preferable but ASFT remains robust overall. For batch size, we sweep values from 32 to 256. The results show stable performance across the full range, with only small fluctuations. This indicates that batch size is not a sensitive factor for ASFT, and standard settings suffice. Overall, ASFT demonstrates both strong robustness and low sensitivity to key hyperparameters, maintaining a consistent advantage over SFT and DFT in medical QA fine-tuning tasks. The detailed results for learning rate and batch size sweeps are shown in Appendix E.2.

## 6 ANALYSIS AND DISCUSSION

### 6.1 SCALING ANALYSIS ACROSS MODEL SIZE

We evaluate fine-tuning methods on medical-domain datasets using 10k training samples (Figure see 3, detailed results in Appendix E.1). Our experiments cover both LLaMA-2 (7B and 70B) and Qwen2.5 (7B, 32B, and 72B) models. Across all model sizes, ASFT consistently outperforms Base, SFT, and DFT, and its improvements remain robust as models scale, demonstrating effective and stable adaptation under low-resource settings.

| Methods | MedQA | MMLU | MedMCQA | Avg. |
|---|---|---|---|---|
| LLaMA-2-7B | 29.85 | 30.52 | 33.76 | 31.38 |
| *Supervised Fine-Tuning* | | | | |
| DFT | 24.67 ↓5.18 | 22.82 ↓7.70 | 30.43 ↓3.33 | 25.97 ↓5.41 |
| iw-SFT | 28.36 ↑1.49 | 35.89 ↑5.37 | 34.88 ↑1.12 | 33.04 ↑1.66 |
| SFT | 33.31 ↑3.56 | 33.52 ↑3.00 | 33.28 ↓0.48 | 33.44 ↑2.06 |
| **ASFT** | **39.28** ↑9.43 | **46.37** ↑15.85 | **40.45** ↑6.69 | **42.03** ↑10.65 |
| *Reinforcement Learning* | | | | |
| GRPO | 30.48 ↑0.63 | 32.46 ↑1.94 | 34.64 ↑0.88 | 32.53 ↑1.15 |
| DAPO | 39.75 ↑9.90 | 48.63 ↑18.11 | 38.37 ↑4.61 | 42.25 ↑10.87 |
| *Continual RL* | | | | |
| SFT + DAPO | 36.84 ↑6.99 | 44.76 ↑14.24 | 39.11 ↑5.35 | 40.24 ↑8.86 |
| **ASFT + DAPO** | **41.32** ↑11.47 | **49.54** ↑19.02 | **41.45** ↑7.69 | **44.10** ↑12.72 |

Table 2: Comparison of different post-training strategies on medical benchmarks. **Bold** numbers indicate the best performance in each group, and rows with blue background highlight our ASFT approach.

## 6.2 COMPARISON WITH REINFORCEMENT LEARNING METHODS

Table 2 presents the performance comparison of different fine-tuning methods on several medical reasoning benchmarks. Our proposed method, **ASFT**, consistently outperforms all SFT-based approaches, including standard SFT, SFT with KL regularization, DFT, and iw-SFT, demonstrating the effectiveness of our adaptive weighting strategy. For example, ASFT achieves an average score of 42.03, which is substantially higher than iw-SFT (33.04) and SFT (33.44). However, as expected, ASFT still falls short of advanced reinforcement learning-based methods such as DAPO, which achieves an average of 42.25, indicating that while our method narrows the gap with RL approaches, RL still maintains a slight advantage in these tasks. The experimental results validate our theoretical analysis from Section 4. The comparison between SFT and DFT (33.44 vs 25.97) confirms DFT's distributional drift problem in knowledge-intensive tasks, while SFT vs ASFT (33.44 vs 42.03) demonstrates the effectiveness of our anchored auxiliary distribution construction. Using final accuracy as a proxy for bound tightness, the performance ordering SFT < ASFT < DAPO (33.44 < 42.03 < 42.25) empirically supports our reward-weighted regression framework, showing that ASFT achieves a tighter lower bound on the RL objective as proven in Theorem 1. The substantial improvement of ASFT over standard SFT while remaining computationally efficient demonstrates the practical value of our theoretically grounded approach.

## 6.3 ASFT AS ENHANCED INITIALIZATION FOR REINFORCEMENT LEARNING

The results in Table 2 further demonstrate that ASFT provides a superior initialization point for subsequent RL fine-tuning. Starting from ASFT and continuing training with DAPO yields consistent gains over SFT + DAPO (44.10 vs. 40.24 average, +3.86 points), with the largest improvements on MMLU-medical (+4.78) and MedQA (+4.48). ASFT + DAPO also surpasses standalone DAPO (44.10 vs. 42.25), indicating that KL-anchored fine-tuning not only improves direct supervised performance but also creates a more stable policy foundation for RL optimization. This finding suggests that the distributional stability provided by ASFT's KL anchoring not only improves direct fine-tuning performance but also creates a more robust foundation for advanced RL algorithms, extending the practical utility of our method beyond standalone applications.

## 6.4 CROSS-DOMAIN VALIDATION ON CODE GENERATION

To examine the generality of ASFT, we fine-tune LLaMA-2-7B on 10k samples from the Magicoder-Evol-Instruct-110K (Wei et al., 2023) dataset using the same setup as Section 5, but for 2 epochs. Evaluation is performed with the evalplus framework (Liu et al., 2023) on HumanEval, HumanEval+ (Chen et al., 2021), MBPP, and MBPP+ (Austin et al., 2021). The results are summarized in Table 3, ASFT achieves the highest average score, notably improving HumanEval and

| Methods | HumanEval | HumanEval+ | MBPP | MBPP+ | Avg. |
|---------|-----------|------------|------|-------|------|
| LLaMA-2-7B | 17.1 | 14.6 | 28.3 | 21.2 | 20.3 |
| SFT | 23.2 ↑6.1 | 19.5 ↑4.9 | **34.1** ↑5.8 | **28.6** ↑7.4 | 26.4 ↑6.1 |
| iw-SFT | 23.8 ↑6.7 | 21.3 ↑6.7 | 31.0 ↑2.7 | 26.5 ↑5.3 | 25.7 ↑5.4 |
| DFT | 15.9 ↓1.2 | 12.8 ↓1.8 | 28.3 ±0.0 | 22.3 ↑1.1 | 19.8 ↓0.5 |
| **ASFT** | **27.2** ↑10.1 | **21.4** ↑6.8 | 32.5 ↑4.2 | 26.7 ↑5.5 | **27.0** ↑6.7 |

Table 3: Performance (%) on code generation with LLaMA-2-7B. **Bold** numbers indicate the best performance in each column, and rows with blue background highlight our ASFT approach.

HumanEval+ while remaining competitive on MBPP, confirming that the anchoring mechanism generalizes effectively to code generation.

## 6.5 COMPUTATIONAL EFFICIENCY AND MEMORY ANALYSIS

While ASFT demonstrates superior performance across both knowledge-intensive and reasoning-intensive domains, the KL divergence computation against the reference model introduces significant computational overhead that merits careful analysis. During training, ASFT requires maintaining the reference model $\pi_{\text{base}}$ in memory alongside the training model $\pi_\theta$, effectively doubling GPU memory consumption from 38.96GB to 88.02GB for full-parameter fine-tuning on LLaMA-2-7B.

The KL computation additionally introduces approximately 23.7% training time overhead compared to standard SFT (0.648 vs 0.524 hours), though remaining substantially more efficient than full RL approaches like GRPO (51.24 hours). All training times are measured on a single NVIDIA A100 GPU. More critically for deployment, inference requires loading both models simultaneously, creating scalability concerns where memory efficiency is paramount.

| Method | Time (hrs) | Memory (GB) | Acc |
|--------|-----------|-------------|-----|
| SFT | 0.524 | 38.96 | 33.04 |
| DFT | **0.521** | **38.90** | 25.97 |
| iw-SFT | 8.287 | 100.05 | 33.04 |
| GRPO | 51.24 | 483.98 | 32.53 |
| DAPO | 21.595 | 488.26 | **42.25** |
| ASFT-LoRA | **0.594** | **40.70** | 39.45 |
| ASFT | 0.648 | 88.02 | **42.03** |

Table 4: Training time, memory, and accuracy of different methods on LLaMA-2-7B.

To address these practical limitations, we propose ASFT-LoRA, leveraging the mathematical properties of low-rank adaptation to enable memory-efficient implementation. The key insight exploits LoRA's parameter decomposition $\Delta W = BA$ where the fine-tuned model becomes $\pi_\theta(y|x) = \pi_{\text{base}}(y|x; W_{\text{base}} + BA)$. This parameterization enables computing $D_{\text{KL}}(\pi_{\text{base}}(\cdot|s)\|\pi_\theta(\cdot|s))$ using a single model instance by dynamically switching between $W_{\text{base}}$ and $W_{\text{base}} + BA$ during forward passes, eliminating the need for separate model copies while preserving theoretical anchoring guarantees. As shown in Table 4, ASFT-LoRA (rank=8, lr=5e-4) reduces memory consumption to 40.70GB (comparable to standard SFT) and training time to 0.594 hours (13.4% overhead over SFT), while achieving substantial improvements over SFT baselines (39.45 vs 33.04 overall accuracy). Notably, ASFT-LoRA with appropriate learning rate selection achieves 93.9% of full-parameter ASFT's performance (39.45 vs 42.03 overall accuracy) while requiring less than half the memory, demonstrating the versatility of our anchoring framework across different parameter-efficiency regimes. We observe that LoRA-based training benefits from higher learning rates (5e-4) compared to full fine-tuning (2e-5), consistent with recent findings on low-rank adaptation (Hu et al., 2021).

## 6.6 ALTERNATIVE ANCHORING MECHANISMS.

Our theoretical framework reveals that the key to stable optimization lies in providing distributional anchoring to prevent drift, and in principle, any regularization term that constrains the policy distribution can serve this purpose. While KL divergence provides effective anchoring, we investigate whether other terms—particularly the standard SFT loss itself—can offer a more lightweight alternative. The SFT loss $\mathcal{L}_{\text{SFT}}(\theta) = -\mathbb{E}_{(x,y^*)\sim\mathcal{D}}[\log \pi_\theta(y^* \mid x)]$ naturally encourages the model to stay close to demonstrated behaviors, suggesting it may provide implicit anchoring effects as observed in Figure 1.

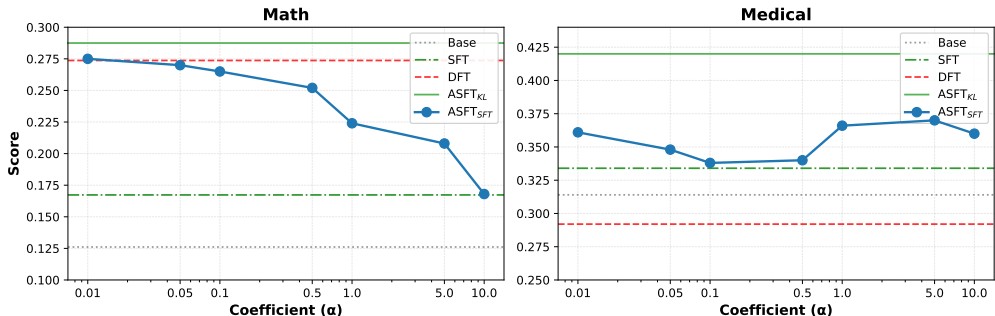

Figure 4: Comparison of KL-based anchoring ($\text{ASFT}_{KL}$) versus SFT loss-based anchoring ($\text{ASFT}_{SFT}$) across different coefficient values ($\alpha$ or $\lambda$) on math reasoning and medical knowledge tasks. $\text{ASFT}_{KL}$ maintains stable performance across all values, while $\text{ASFT}_{SFT}$ shows domain-specific sensitivity: degradation at large coefficients for math tasks and volatility for medical tasks.

We therefore explore a hybrid objective that combines DFT's adaptive reweighting with SFT-based anchoring:

$$\mathcal{L}_{\text{ASFT}-\text{SFT}}(\theta) = \mathcal{L}_{\text{DFT}}(\theta) + \alpha \mathcal{L}_{\text{SFT}}(\theta) \tag{7}$$

where $\alpha$ controls the anchoring strength. Compared to KL-based anchoring, this approach is computationally more efficient as it does not require maintaining a separate reference model during training. As shown in Figure 4, we conduct a grid search over $\alpha \in \{0.01, 0.05, 0.1, 0.5, 1.0, 5.0, 10.0\}$ on both math and medical tasks.

Interestingly, we observe domain-specific patterns: on mathematical reasoning tasks, ASFT-SFT achieves competitive performance at moderate $\alpha$ values (0.05-0.5), but degrades significantly at larger coefficients (dropping from 0.275 to 0.168 as $\alpha$ increases from 0.01 to 10.0). This suggests that for reasoning tasks where the base model already possesses strong prior knowledge, moderate SFT anchoring can effectively stabilize training by activating these capabilities without over-constraining exploration. In contrast, on medical knowledge tasks, ASFT-SFT shows more volatile behavior across different $\alpha$ values, with performance fluctuating between 0.340 and 0.370. This instability likely reflects the challenge of balancing knowledge injection (requiring departure from the base distribution) with stability preservation in knowledge-intensive domains. Throughout both settings, ASFT with KL anchoring (green lines) maintains consistently superior and stable performance across all coefficient values, demonstrating that while SFT loss provides useful anchoring, KL divergence offers more robust distributional control, particularly for tasks requiring significant distribution shift from the base model.

# 7 CONCLUSION

We present Anchored Supervised Fine-Tuning (ASFT), a principled approach that addresses the fundamental trade-off between supervised fine-tuning's efficiency and reinforcement learning's generalization. By grounding Dynamic Fine-Tuning in the reward-weighted regression framework, we show that DFT achieves tighter RL lower bounds than SFT but suffers from distributional drift. ASFT resolves this through lightweight KL anchoring, preserving tightness while ensuring stability. Empirically, ASFT consistently outperforms SFT and DFT across mathematical reasoning, medical knowledge injection, and code generation. The method achieves substantial improvements with minimal computational overhead, demonstrating that principled theoretical analysis can lead to both stronger guarantees and practical gains.

ACKNOWLEDGEMENTS

This project was supported by National Natural Science Foundation of China (No. 62306132), Guangdong Basic and Applied Basic Research Foundation (No. 2025A1515011564). We thank the anonymous reviewers for their insightful feedback on this work.

ETHICS STATEMENT

All experiments in this work were conducted using publicly available datasets and standard benchmarks. The proposed Anchored Supervised Fine-Tuning (ASFT) method is intended for research purposes to improve model generalization and stability. When deploying models trained with ASFT in high-stakes applications, we recommend thorough validation, human oversight, and appropriate safeguards to ensure outputs are accurate, reliable, and aligned with ethical guidelines. We acknowledge potential risks of harmful or biased content and encourage the use of input filtering, output moderation, and ongoing monitoring to mitigate such risks. Our goal is to contribute to robust, efficient, and trustworthy AI systems that can be safely deployed.

REPRODUCIBILITY STATEMENT

We provide comprehensive implementation details in Section 5, including specific hyperparameters, datasets, and evaluation protocols. All experiments use publicly available datasets and standard benchmarks. We have released our implementation code to ensure reproducibility of our results. The code is available at `https://github.com/zhuchichi56/ASFT`.

LIMITATIONS

We evaluate ASFT on three domains—mathematical reasoning, medical knowledge, and code generation. Although results are consistent across these areas, broader evaluation across diverse task types and domains is needed to fully establish the method's generalizability.

Our theoretical analysis shows that DFT's effectiveness is fundamentally dependent on the prior knowledge embedded in the base model, which accounts for its strong performance in reasoning tasks with well-established priors, but weaker results in knowledge-intensive domains requiring more substantial distributional shifts. Although our KL-based anchoring mitigates DFT's instability, the method's reliance on model priors persists. Our ablation study on alternative anchoring mechanisms (Section 6.6) suggests that domain-adaptive anchoring strategies could enable more robust generalization across domains; however, developing a systematic framework for such adaptation remains an open challenge.

Additionally, ASFT incurs higher computational costs than standard SFT due to the KL divergence computation, requiring doubled GPU memory (from 38.96GB to 88.02GB for LLaMA-2-7B) and 23.7% training time overhead. Our ASFT-LoRA variant (Section 6.5) provides a practical solution by reducing memory to 40.70GB with only 7.3% overhead, though with some performance trade-off.

Our work lacks systematic analysis of what capabilities or behaviors SFT may lose when transitioning to ASFT. While we observe consistent improvements, a more comprehensive understanding of the trade-offs involved in our reweighting scheme would provide valuable insights for practitioners.

FUTURE WORK

A promising direction for future research is extending ASFT to continual learning settings, where models must adapt to new tasks while preserving previously acquired knowledge. The anchoring mechanism in ASFT naturally aligns with continual learning goals: the KL regularization that prevents distributional drift during single-task fine-tuning could be adapted to prevent catastrophic forgetting when learning sequential tasks. By maintaining anchors to both the base model and

checkpoints from previous tasks, ASFT's framework could potentially enable stable, progressive capability acquisition across multiple domains.

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

## A   LLM USAGE

In the preparation of this paper, we only used large language models (LLMs) as an assistive tool for grammar correction and text polishing.

## B    LIMITATIONS

We evaluate ASFT on three domains—mathematical reasoning, medical knowledge, and code generation. Although results are consistent across these areas, broader evaluation across diverse task types and domains is needed to fully establish the method's generalizability.

Our work lacks systematic analysis of what capabilities or behaviors SFT may lose when transitioning to ASFT. While we observe consistent improvements, a more comprehensive understanding of the trade-offs involved in our reweighting scheme would provide valuable insights for practitioners.

## C    BROADER IMPACTS

ASFT offers a lightweight and computationally efficient alternative to standard supervised fine-tuning, requiring only KL regularization rather than complex reward modeling. This simplicity makes advanced fine-tuning techniques more accessible to researchers and practitioners with limited computational resources. The method shows potential as a practical path forward for improving language model post-training, particularly in specialized domains requiring both knowledge injection and reasoning capabilities. However, as with any fine-tuning approach, careful evaluation and validation remain essential when deploying models in high-stakes applications such as medical or educational settings.

## D    THEORETICAL FOUNDATIONS

This appendix provides detailed derivations for the key theoretical results presented in Section 3.

### D.1    DERIVATION OF SFT AS RL LOWER BOUND

We provide a complete proof of Proposition 1, following the theoretical framework established in (Qin & Springenberg, 2025), showing that SFT optimizes a lower bound on the RL objective in sparse reward settings.

*Proof of Proposition 1.* We start with the RL objective under sparse rewards $R(\tau) = \mathbb{I}[y = y^*]$:

$$J(\theta) = \mathbb{E}_{\tau \sim \pi_\theta}[R(\tau)] = \mathbb{E}_{\tau \sim \pi_\theta}[\mathbb{I}[y = y^*]] \tag{8}$$

Since we only observe trajectories from the reference distribution $\pi_{\text{ref}}$, we use importance sampling to rewrite this expectation. Under the assumption $\text{supp}(\pi_\theta) \subseteq \text{supp}(\pi_{\text{ref}})$:

$$J(\theta) = \mathbb{E}_{\tau \sim \pi_{\text{ref}}}\left[\frac{\pi_\theta(\tau)}{\pi_{\text{ref}}(\tau)}\mathbb{I}[y = y^*]\right] \tag{9}$$

Now we apply the fundamental inequality $u \geq 1 + \log u$ for all $u > 0$. Setting $u = \frac{\pi_\theta(\tau)}{\pi_{\text{ref}}(\tau)}$:

$$\frac{\pi_\theta(\tau)}{\pi_{\text{ref}}(\tau)} \geq 1 + \log \frac{\pi_\theta(\tau)}{\pi_{\text{ref}}(\tau)} = 1 + \log \pi_\theta(\tau) - \log \pi_{\text{ref}}(\tau) \tag{10}$$

Substituting this into our importance-sampled expression:

$$J(\theta) \geq \mathbb{E}_{\tau \sim \pi_{\text{ref}}}\left[(1 + \log \pi_\theta(\tau) - \log \pi_{\text{ref}}(\tau))\,\mathbb{I}[y = y^*]\right] \tag{11}$$
$$= \mathbb{E}_{\tau \sim \pi_{\text{ref}}}[\mathbb{I}[y = y^*]] + \mathbb{E}_{\tau \sim \pi_{\text{ref}}}[\mathbb{I}[y = y^*]\log \pi_\theta(\tau)] \tag{12}$$
$$- \mathbb{E}_{\tau \sim \pi_{\text{ref}}}[\mathbb{I}[y = y^*]\log \pi_{\text{ref}}(\tau)] \tag{13}$$

The first and third terms are constants independent of $\theta$. Let $c_{\text{ref}} = \mathbb{E}_{\tau \sim \pi_{\text{ref}}}[\mathbb{I}[y = y^*]] = \mathbb{P}_{\pi_{\text{ref}}}(\tau \in D^+)$. Then:

$$J(\theta) \geq c_{\text{ref}} + \mathbb{E}_{\tau \sim \pi_{\text{ref}}}[\mathbb{I}[y = y^*]\log \pi_\theta(\tau)] + \text{const} \tag{14}$$

The expectation over indicator-weighted trajectories can be rewritten as an expectation over the filtered dataset $D^+ = \{(x, y^*) \mid R(x, y^*) = 1\}$:

$$\mathbb{E}_{\tau \sim \pi_{\text{ref}}}[\mathbb{I}[y = y^*] \log \pi_\theta(\tau)] = c_{\text{ref}} \mathbb{E}_{\tau \in D^+}[\log \pi_\theta(\tau)] \tag{15}$$

Dropping the constant terms, we obtain:

$$J(\theta) \geq c_{\text{ref}} \mathbb{E}_{\tau \in D^+}[\log \pi_\theta(\tau)] \tag{16}$$

This completes the proof. Note that the right-hand side is precisely the SFT objective (up to scaling), establishing that SFT optimizes a lower bound on the RL objective. $\square$

## D.2 FROZEN AUXILIARY DISTRIBUTION AND OPTIMIZATION VALIDITY

We formalize how DFT's stop-gradient mechanism ensures valid lower bound optimization by treating the auxiliary distribution as constant during each update step.

**Lemma 1** (Frozen-q Surrogate). *At iteration $k$, define the auxiliary distribution*

$$q_k(\tau) \propto \pi_{\text{ref}}(\tau \mid D^+) \, p_{\theta_k}(\tau) \tag{17}$$

*where $\theta_k$ is the current parameter. Then optimizing*

$$\max_\theta \mathbb{E}_{\tau \in D^+}\left[\frac{q_k(\tau)}{\pi_{\text{ref}}(\tau)} \log \pi_\theta(\tau)\right] \tag{18}$$

*with $q_k$ treated as constant (independent of $\theta$) yields a valid lower bound on $J(\theta)$ that differs only by a $\theta$-independent constant.*

*Proof.* Following the derivation in Appendix D.3, we have:

$$J(\theta) \geq \mathbb{E}_{\tau \sim \pi_{\text{ref}}}\left[\frac{q_k(\tau)}{\pi_{\text{ref}}(\tau)} R(\tau)\right] + \mathbb{E}_{\tau \sim \pi_{\text{ref}}}\left[\frac{q_k(\tau)}{\pi_{\text{ref}}(\tau)} R(\tau) \log \pi_\theta(\tau)\right]$$
$$- \mathbb{E}_{\tau \sim \pi_{\text{ref}}}\left[\frac{q_k(\tau)}{\pi_{\text{ref}}(\tau)} R(\tau) \log q_k(\tau)\right] \tag{19}$$

When $q_k$ is constructed from frozen parameter $\theta_k$ (via stop-gradient), the first and third terms become constants independent of the optimization variable $\theta$. Therefore, maximizing the second term is equivalent to maximizing a valid lower bound on $J(\theta)$.

The stop-gradient operator $\text{sg}[\cdot]$ in DFT implements this frozen-$q$ mechanism: at each iteration, $q_k$ depends on $\theta_k$ from the previous step, but is treated as fixed when computing gradients with respect to the current $\theta$. $\square$

This formalization clarifies that the "dropped constant terms" in our derivations are legitimate because they are genuinely $\theta$-independent within each optimization iteration, making DFT a valid iterative lower bound maximization procedure.

## D.3 FROM IMPORTANCE SAMPLING TO IMPORTANCE-WEIGHTED LOWER BOUNDS

We now derive the generalized importance-weighted framework that enables tighter bounds through auxiliary distributions.

Starting from the importance-sampled RL objective:

$$J(\theta) = \mathbb{E}_{\tau \sim \pi_{\text{ref}}}\left[\frac{\pi_\theta(\tau)}{\pi_{\text{ref}}(\tau)} R(\tau)\right] \tag{20}$$

We introduce an auxiliary distribution $q(\tau)$ with $\text{supp}(q) \supseteq \text{supp}(\pi_\theta)$ and rewrite the importance ratio:

$$J(\theta) = \mathbb{E}_{\tau \sim \pi_{\text{ref}}}\left[\frac{q(\tau)}{\pi_{\text{ref}}(\tau)} \cdot \frac{\pi_\theta(\tau)}{q(\tau)} R(\tau)\right] \tag{21}$$

Now we apply the inequality $u \geq 1 + \log u$ to the ratio $\frac{\pi_\theta(\tau)}{q(\tau)}$:

$$\frac{\pi_\theta(\tau)}{q(\tau)} \geq 1 + \log \frac{\pi_\theta(\tau)}{q(\tau)} = 1 + \log \pi_\theta(\tau) - \log q(\tau) \tag{22}$$

Substituting this bound:

$$J(\theta) \geq \mathbb{E}_{\tau \sim \pi_{\text{ref}}} \left[ \frac{q(\tau)}{\pi_{\text{ref}}(\tau)} \left(1 + \log \pi_\theta(\tau) - \log q(\tau)\right) R(\tau) \right] \tag{23}$$

$$= \mathbb{E}_{\tau \sim \pi_{\text{ref}}} \left[ \frac{q(\tau)}{\pi_{\text{ref}}(\tau)} R(\tau) \right] + \mathbb{E}_{\tau \sim \pi_{\text{ref}}} \left[ \frac{q(\tau)}{\pi_{\text{ref}}(\tau)} R(\tau) \log \pi_\theta(\tau) \right] \tag{24}$$

$$- \mathbb{E}_{\tau \sim \pi_{\text{ref}}} \left[ \frac{q(\tau)}{\pi_{\text{ref}}(\tau)} R(\tau) \log q(\tau) \right] \tag{25}$$

When treating $q$ as fixed (independent of $\theta$, see Lemma 1), the first and third terms are constants, so we can drop them from the optimization objective:

$$J(\theta) \geq \mathbb{E}_{\tau \sim \pi_{\text{ref}}} \left[ \frac{q(\tau)}{\pi_{\text{ref}}(\tau)} R(\tau) \log \pi_\theta(\tau) \right] + \text{const} \tag{26}$$

For sparse rewards $R(\tau) = \mathbb{I}[y = y^*]$, this reduces to:

$$J(\theta) \geq \mathbb{E}_{\tau \sim \pi_{\text{ref}}} \left[ \frac{q(\tau)}{\pi_{\text{ref}}(\tau)} \mathbb{I}[y = y^*] \log \pi_\theta(\tau) \right] + \text{const} \tag{27}$$

Converting to an expectation over the filtered dataset $D^+$:

$$J(\theta) \geq c_{\text{ref}} \mathbb{E}_{\tau \in D^+} \left[ \frac{q(\tau)}{\pi_{\text{ref}}(\tau)} \log \pi_\theta(\tau) \right] + \text{const} \tag{28}$$

where $c_{\text{ref}} = \mathbb{P}_{\pi_{\text{ref}}}(\tau \in D^+)$.

**Key Insights:**

1. When $q(\tau) = \pi_{\text{ref}}(\tau)$, we recover the standard SFT bound from Proposition 1.

2. As $q(\tau) \to \pi_\theta(\tau)$, the bound becomes tighter since the inequality $\frac{\pi_\theta(\tau)}{q(\tau)} \geq 1 + \log \frac{\pi_\theta(\tau)}{q(\tau)}$ approaches equality.

3. The choice of $q$ involves a fundamental trade-off: tighter bounds (by making $q$ closer to $\pi_\theta$) versus stability (by keeping $q$ close to $\pi_{\text{ref}}$).

This theoretical framework provides the foundation for both existing methods like DFT and our proposed ASFT approach, which aims to achieve tight bounds while maintaining optimization stability through principled anchoring mechanisms.

### D.4   PROOF OF TIGHTNESS

We provide the detailed proof that the DFT auxiliary distribution $q$ yields a strictly tighter lower bound than standard SFT whenever the policy distribution is non-degenerate.

*Proof of Theorem 1.* Let $X = p_\theta(\tau)$ with $\tau \sim \pi_{\text{ref}}(\cdot \mid D^+)$, and $f(x) = \log x$. We compare:

$$B_{\text{SFT}} = c_{\text{ref}} \mathbb{E}[f(X)], \qquad B_{\text{DFT}} = c_{\text{ref}} \frac{\mathbb{E}[X f(X)]}{\mathbb{E}[X]} \tag{29}$$

The difference is:

$$B_{\text{DFT}} - B_{\text{SFT}} = c_{\text{ref}} \left( \frac{\mathbb{E}[X f(X)]}{\mathbb{E}[X]} - \mathbb{E}[f(X)] \right) \tag{30}$$

$$= \frac{c_{\text{ref}}}{\mathbb{E}[X]} \left( \mathbb{E}[X f(X)] - \mathbb{E}[X] \mathbb{E}[f(X)] \right) \tag{31}$$

$$= \frac{c_{\text{ref}}}{\mathbb{E}[X]} \text{Cov}(X, f(X)) \tag{32}$$

Since $f(x) = \log x$ is strictly increasing on $(0, 1]$, variables $X$ and $f(X)$ are comonotone, yielding $\text{Cov}(X, f(X)) \geq 0$ with equality iff $X$ is constant. Therefore $B_{\text{DFT}} \geq B_{\text{SFT}}$, with strict inequality when $\text{Var}(X) > 0$. $\qquad\square$

# E  EXPERIMENTS

## E.1  MODEL SCALE RESULTS

| Model | Methods | MedQA | MMLU | MedMCQA | Avg. |
|-------|---------|-------|------|---------|------|
| LLaMA-2-7B | Base | 29.85 | 30.52 | 33.76 | 31.38 |
| | SFT | 33.31 ↑3.46 | 33.52 ↑3.00 | 33.28 ↓0.48 | 33.37 ↑1.99 |
| | SFT w KL | 29.22 ↓0.63 | 30.63 ↑0.11 | 33.01 ↓0.75 | 30.95 ↓0.43 |
| | iw-SFT | 35.35 ↑5.50 | 38.92 ↑8.40 | 34.74 ↑0.98 | 36.34 ↑4.96 |
| | DFT | 29.69 ↓0.16 | 26.69 ↓3.83 | 31.20 ↓2.56 | 29.19 ↓2.19 |
| | GRPO | 30.48 ↑0.63 | 32.46 ↑1.94 | 34.64 ↑0.88 | 32.53 ↑1.15 |
| | DAPO | **39.75** ↑9.90 | **48.63** ↑18.11 | 38.37 ↑4.61 | **42.25** ↑10.87 |
| | **ASFT** | 39.28 ↑9.43 | 46.37 ↑15.85 | **40.45** ↑6.69 | 42.03 ↑10.65 |
| LLaMA-2-70B | Base | 47.37 | 65.32 | 47.21 | 53.30 |
| | DFT | 36.84 ↓10.53 | 46.77 ↓18.55 | 42.39 ↓4.82 | 42.00 ↓11.30 |
| | SFT | 41.24 ↓6.13 | 47.02 ↓18.30 | 36.43 ↓10.78 | 41.56 ↓11.74 |
| | **ASFT** | **49.57** ↑2.20 | **65.86** ↑0.54 | **50.61** ↑3.40 | **55.35** ↑2.05 |
| Qwen2.5-7B | Base | 62.53 | 74.84 | 58.45 | 65.27 |
| | SFT | 59.07 ↓3.46 | 71.19 ↓3.65 | 49.80 ↓8.65 | 60.02 ↓5.25 |
| | SFT w KL | 61.12 ↓1.41 | 69.77 ↓5.07 | 51.52 ↓6.93 | 60.80 ↓4.47 |
| | DFT | 36.45 ↓26.08 | 54.65 ↓20.19 | 45.90 ↓12.55 | 45.67 ↓19.60 |
| | GRPO | **63.00** ↑0.47 | 76.16 ↑1.32 | 58.93 ↑0.48 | 66.03 ↑0.76 |
| | DAPO | 63.94 ↑1.41 | 73.82 ↓1.02 | 58.88 ↑0.43 | 65.55 ↑0.28 |
| | **ASFT** | 61.98 ↓0.55 | **76.60** ↑1.76 | **59.00** ↑0.55 | **65.86** ↑0.59 |
| Qwen2.5-32B | Base | 61.90 | 62.50 | 45.09 | 56.50 |
| | SFT | 67.09 ↑5.19 | 76.12 ↑13.62 | 61.20 ↑16.11 | 68.14 ↑11.64 |
| | DFT | 68.42 ↑6.52 | 57.03 ↓5.47 | 42.51 ↓2.58 | 55.99 ↓0.51 |
| | **ASFT** | **71.80** ↑9.90 | **79.23** ↑16.73 | **58.76** ↑13.67 | **69.93** ↑13.43 |
| Qwen2.5-72B | Base | 75.57 | 78.28 | 64.91 | 72.92 |
| | SFT | 28.04 ↓47.53 | 34.57 ↓43.71 | 38.92 ↓25.99 | 33.84 ↓39.08 |
| | DFT | 71.01 ↓4.56 | 79.04 ↑0.76 | 64.69 ↓0.22 | 71.58 ↓1.34 |
| | **ASFT** | **77.69** ↑2.12 | **87.08** ↑8.80 | **71.00** ↑6.09 | **78.59** ↑5.67 |

Table 5: Performance of various fine-tuning methods on medical domain datasets for both LLaMA-2 and Qwen2.5 series. **Bold** numbers indicate the best performance in each group, and rows with  blue background  highlight our ASFT approach. Arrows with ↑ and ↓ indicate performance improvements and degradations relative to the base model. All values are reported in percentages.

## E.2   Training Hyper-Parameters

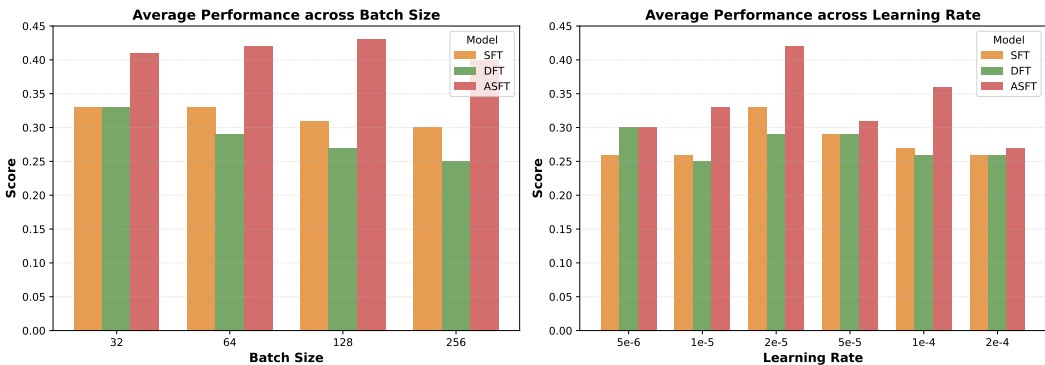

Figure 5: Ablation study of ASFT on MedCAQA (10k) using LLaMA-2-7B. **Right:** Effect of batch size, indicating stable performance across a wide range (32–256). ASFT consistently outperforms SFT and DFT under all settings. **Left:** Effect of different learning rates on average performance, showing that intermediate rates (1e-5 and 1e-4) achieve the best results.

## F  DATASET EXAMPLE

### F.1  MEDICAL KNOWLEDGE DATASETS

---

**Medical Knowledge Training Set**

**Instruction:**
Disulfiram is a type of:-
A. Aversion therapy
B. Anticraving therapy
C. Detoxification
D. Opioid management therapy

**Response:**
Deterrent agents/Aversive agents Disulfiram (tetraethyl thiuram disulfide) When alcohol is ingested by a person who is on disulfiram, alcohol-derived acetaldehyde cannot be oxidized to acetate and this leads to an accumulation of acetaldehyde in blood. This causes the impoant disulfiram-ethanol reaction (DER) characterized by flushing, tachycardia, hypotension, tachypnoea, palpitations, headache, sweating, nausea, vomiting, giddiness and a sense of impending doom associated with severe anxiety. When we try to decrease the frequency of behavior because it can cause a really bad consequence It will be called as an aversion or Aversive therapy. Anti-craving agents Acamprosate, naltrexone and SSRIs (such as fluoxetine) are among the medications tried as anti-craving agents in alcohol dependence. Detoxification Detoxification is the treatment of alcohol withdrawal symptoms, i.e. symptoms produced by the removal of the 'toxin' (alcohol). The drugs of choice for detoxification are usually benzodiazepines.
So the answer to this question is A

---

Figure 6: An example from the medical knowledge training set illustrating an instruction-response pair on alcohol dependence therapy

---

**Medical Knowledge Test Set**

**Question:**
Which of the following is not true for myelinated nerve fibers:

**Options:**
A. Impulse through myelinated fibers is slower than non-myelinated fibers
B. Membrane currents are generated at nodes of Ranvier
C. Saltatory conduction of impulses is seen
D. Local anesthesia is effective only when the nerve is not covered by myelin sheath

**Answer:** A

---

Figure 7: An example from the medical knowledge test set illustrating a multiple-choice question on physiology (myelinated nerve fibers)

## F.2 MATH REASONING DATASETS

---

**Math Reasoning Training Set**

**Instruction:**
Let $a$, $b$, and $c$ be the roots of the equation $x^3 - 2x - 5 = 0$. Find $\frac{1}{a-2} + \frac{1}{b-2} + \frac{1}{c-2}$.

**Response:**
First, substitute $x = z + 2$ to transform the polynomial so that $z + 2$ are the new roots, i.e., $a - 2$, $b - 2$, and $c - 2$ are the roots for $z$. We then have:

$$(z+2)^3 - 2(z+2) - 5 = z^3 + 6z^2 + 12z + 8 - 2z - 4 - 5 = z^3 + 6z^2 + 10z - 1$$

We are interested in $\frac{1}{a-2} + \frac{1}{b-2} + \frac{1}{c-2}$, which are the reciprocals of the roots of $z^3 + 6z^2 + 10z - 1$. By Vieta's formulas, the sum of the reciprocals of the roots of a polynomial $z^3 + pz^2 + qz + r$ is $-q/r$. For our polynomial, $p = 6$, $q = 10$, and $r = -1$, so:

$$\frac{1}{a-2} + \frac{1}{b-2} + \frac{1}{c-2} = -\frac{q}{r} = -\frac{10}{-1} = \boxed{10}$$

---

Figure 8: An example from the math reasoning training set illustrating an instruction-response pair on algebraic root manipulation

---

**Math Reasoning Test Set**

**Question:**
What is the smallest positive perfect cube that can be written as the sum of three consecutive integers?

**Answer:**
27

**Solution:**
The sum of three consecutive integers takes the form $(k - 1) + (k) + (k + 1) = 3k$ and hence is a multiple of 3. Conversely, if a number $n$ is a multiple of 3, then $n/3 - 1$, $n/3$, and $n/3 + 1$ are three consecutive integers that sum to give $n$. Therefore, a number is a sum of three consecutive integers if and only if it is a multiple of 3. The smallest positive perfect cube that is a multiple of 3 is $3^3 = \boxed{27}$.

---

Figure 9: An example from the math reasoning test set illustrating a problem with its solution on sums of consecutive integers

