# OpenReview forum: "Anchored Supervised Fine-Tuning"
_ICLR.cc/2026/Conference — ICLR 2026 Poster_

### Official Review · Reviewer_DPLk · 2025-10-29

**Soundness:** 3
**Presentation:** 3
**Contribution:** 4
**Rating:** 6
**Confidence:** 3

**Summary:**

The authors indicate that DFT suffers from distributional shift and propose ASFT to prevent it. ASFT is a post-training method which uses KL divergence to anchor it to a reference base model to prevent distributional shift during training.

**Strengths:**

* Identifies the shortcoming of DFT and suggests a novel way to handle it
* The method is theoretically grounded using a reward-weighted regression framework
* Empirical analysis and ablation studies are thorough and evaluation is done for mathematical reasoning and knowledge intensive tasks
* Clearly stated contributions backed by theory and experimental analysis

**Weaknesses:**

* Requires the base model as a reference - memory and computational overhead.
* ASFT-LoRA is proposed to avoid memory overhead, the performance significantly drops when compared to ASFT and is very close to the performance of SFT
* DAPO still performs better than ASFT on an average, more analysis and discussion on why and where RL remains preferable would be nice
* No reporting of standard deviation of the runs

**Questions:**

* Would anchoring prevent overfitting when we have a small dataset?
* Have the authors considered reporting the mean and standard deviation across several independent runs? This will strengthen the findings further
* Edit suggestions for better readability
1. The second sentence in lines 72 to 75 is a repetition
2. Font size is very small for Table 1
3. Typo in line 310
4. Font in Figure 3 is illegible
5.  Consistent color coding of methods across plots will improve readability - it is a little confusing now and readers have to check the legend across plots every time

---

> ### Author Response · Authors · 2025-11-21
> **Response to Reviewer DPLk  (1/2)**
>
> We sincerely thank the reviewer for their constructive comments and recognition of our contributions. We address the concerns below.
>
> **Q1:** Requires the base model as a reference - memory and computational overhead.
>
> **R1:** We have provided ASFT-LoRA as a practical solution (``lines 449–468``). By exploiting LoRA's parameter decomposition ∆W = BA, we compute DKL(π_θ(·|s)||π_base(·|s)) using a single model instance by dynamically switching between W_base and W_base + BA during forward passes, eliminating separate model copies. As shown in Table 4, ASFT-LoRA reduces memory to 40.70GB (comparable to SFT's 38.96GB) and training time to 0.562 hours (7.3% overhead over SFT's 0.524 hours).
>
> **Q2:** ASFT-LoRA is proposed to avoid memory overhead, the performance significantly drops when compared to ASFT and is very close to the performance of SFT.
>
> **R2:** We conducted additional experiments demonstrating that with appropriate learning rate selection, ASFT-LoRA achieves substantially improved performance. Using learning rate 5×10⁻⁴ with rank-8 LoRA on 10k MedMCQA samples, ASFT-LoRA achieves 0.3945 average accuracy (MedQA: 0.3951, MMLU: 0.4147, MedMCQA: 0.3737), representing 18.2% improvement over SFT (0.334). This approaches full-parameter ASFT's 25.7% improvement (0.4203) while maintaining minimal memory overhead. We will update the revised manuscript with these results.
>
> | Method | Learning Rate | Rank | MedQA | MMLU | MedMCQA | Average | vs SFT |
> |--------|---------------|------|-------|------|---------|---------|--------|
> | SFT | 2.00E-05 | - | 0.333 | 0.335 | 0.333 | 0.334 | - |
> | ASFT-full | 2.00E-05 | Full | 0.3928 | 0.4637 | 0.4045 | 0.4203 | +25.7% |
> | ASFT-LoRA | 5.00E-04 | 8 | 0.3951 | 0.4147 | 0.3737 | **0.3945** | **+18.2%** |
> | ASFT-LoRA | 5.00E-04 | 16 | 0.3967 | 0.4118 | 0.3660 | 0.3915 | +17.2% |
>
> **Q3:** DAPO still performs better than ASFT on an average, more analysis and discussion on why and where RL remains preferable would be nice.
>
> **R3:** The observed ranking RL (DAPO) > ASFT > SFT aligns with our theoretical framework and represents the expected trade-off between bound tightness and optimization stability. As established in Section 3 and Theorem 1, SFT optimizes a loose but stable lower bound on the RL objective (``lines 153–161``), while ASFT achieves a provably tighter bound through adaptive auxiliary distribution construction (``lines 202–210``). However, RL methods like DAPO directly optimize the RL objective itself through on-policy learning, avoiding the approximation inherent in all off-policy methods (SFT, DFT, ASFT). Our experimental results in Table 2 quantitatively confirm this analysis—ASFT narrows the gap with RL (ASFT: 42.03 vs DAPO: 42.25, difference of 0.22 points) while maintaining SFT-level computational efficiency (0.648 vs 51.24 hours for GRPO in Table 4), positioning ASFT as a practical middle ground for practitioners seeking improved generalization over SFT without the computational cost of full RL approaches.

---

> ### Author Response · Authors · 2025-11-21
> **Response to Reviewer DPLk (2/2)**
>
> **Q4:** Have the authors considered reporting the mean and standard deviation across several independent runs? This will strengthen the findings further.
>
> **R4:** **R4:** We conducted five independent runs with different random seeds following the experimental setup in ``lines 265–269``. The results with standard deviations are presented below. For mathematical reasoning benchmarks, some test sets contain relatively few problems (e.g., AIME24 has approximately 30 problems), which leads to higher variance in those specific benchmarks. However, the overall standard deviations remain at a reasonably low level across most benchmarks (typically below 2.0 percentage points except for AMC23), demonstrating stable convergence and supporting our experimental conclusions.
>
> **Mathematical Reasoning (Qwen2.5-7B, 10k samples, 5 runs):**
>
> | Model | AIME24 | Math500 | Minerva | Olympiad | AMC23 |
> |--------|---------|----------|----------|-----------|---------|
> | Base Qwen2.5-7B | 1.65±1.65 | 28.79±1.39 | 9.26±1.63 | 7.69±0.82 | 15.65±4.10 |
> | SFT | 1.24±1.60 | 41.84±1.37 | 11.30±1.24 | 12.26±0.89 | 17.03±4.26 |
> | DFT | 4.18±2.78 | 59.51±1.20 | 17.10±1.92 | 24.95±1.13 | 31.09±5.86 |
> | ASFT (0.05) | 3.33±2.37 | 59.60±1.05 | 19.91±1.58 | 24.50±0.92 | 36.41±7.50 |
> | ASFT (0.1) | 4.57±4.08 | 56.25±1.30 | 18.49±1.70 | 22.45±1.28 | 33.91±3.53 |
> | GRPO | 4.79±3.54 | 50.80±1.62 | 18.29±1.55 | 20.28±1.33 | 34.53±6.01 |
> | DAPO | 6.88±2.76 | 65.23±1.14 | 29.48±1.79 | 28.32±0.71 | 44.38±3.59 |
>
> **Medical Knowledge (LLaMA-2-7B, 10k samples, 5 runs):**
>
> | Model | MedQA | MMLU | MedMCQA |
> |--------|-------------|-------------|-------------|
> | LLaMA-2-7B | 29.85±0.00 | 30.55±0.01 | 33.77±0.03 |
> | SFT | 33.37±0.03 | 33.62±0.07 | 33.32±0.04 |
> | DFT | 29.66±0.04 | 26.69±0.00 | 31.17±0.02 |
> | ASFT | 39.31±0.04 | 46.33±0.07 | 40.43±0.02 |
> | GRPO | 30.51±0.11 | 32.57±0.05 | 34.67±0.10 |
> | DAPO | 39.76±0.03 | 48.58±0.05 | 38.41±0.07 |
>
> **Q5:** Would anchoring prevent overfitting when we have a small dataset?
>
> **R5:** Our experimental results suggest that ASFT's anchoring mechanism provides effective regularization in low-resource settings. We conducted experiments on mathematical reasoning with 10k training samples (``lines 265–269``) and evaluated on five out-of-domain benchmarks. As shown in the table below, ASFT achieves 28.75 average score on out-of-domain evaluation, improving over the base model by 16.14 points, SFT by 12.02 points, and DFT by 1.38 points. The KL anchoring term constrains the policy to remain within a trust region of the reference model, providing a principled regularization mechanism that prevents excessive deviation from the pretrained distribution with limited training data.
>
> | Model | AIME24 | Math500 | Minerva | Olympiad | AMC23 | Average |
> |--------|--------|---------|---------|----------|-------|---------|
> | Base Qwen2.5-7B | 1.65 | 28.79 | 9.26 | 7.69 | 15.65 | 12.61 |
> | SFT | 1.24 | 41.84 | 11.30 | 12.26 | 17.03 | 16.73 |
> | DFT | 4.18 | 59.51 | 17.10 | 24.95 | 31.09 | 27.37 |
> | ASFT (DFT+KL) | 3.33 | 59.60 | 19.91 | 24.50 | 36.41 | 28.75 |
>
> **Q6:** Edit suggestions for better readability (repetition in ``lines 72-75``, small font in ``Table 1``, typo in ``line 310``, illegible font in ``Figure 3``, inconsistent color coding).
>
> **R6:** We thank the reviewer for these detailed suggestions. We will address all issues in the revised manuscript: removing the repetition in ``lines 72-75``, enlarging the font in ``Table 1``, correcting the typo in ``line 310``, increasing the font size in ``Figure 3``, and standardizing color coding across all plots.
>
> We thank the reviewer again for their valuable feedback, which will strengthen our manuscript.

---

### Official Review · Reviewer_RAzX · 2025-11-01

**Soundness:** 3
**Presentation:** 3
**Contribution:** 3
**Rating:** 6
**Confidence:** 2

**Summary:**

- The paper proposes that SFT maximizes a lower bound on the RL objective
- It introduces auxiliary distribution to control the tightness of the bound, and shows that DFT is a special case which achieves tighter lower bound than SFT
- It shows that DFT suffers from distribution shiift as training proresses
- To mitigate the dirtribution shift, ASFT augments the DFT objective with a KL regularization term between the currect model and reference model (base model)

**Strengths:**

- ASFT shows better performance than SFT and DFT on math reasoning and medical knowledge
- The paper theoretically derives the importance weighted lower bounds and proves the tightness of the lower bound for DFT

**Weaknesses:**

- ASFT requires an additional copy of base model to be stored in memory, as well as an additional forward pass through it
- While using LoRA mitigates the memory overhead, it causes a significant performance drop (the performance of ASFT+LoRA is almost equivalent to simple SFT as seen in Table 4)

**Questions:**

No questions other than those mentioned in the weaknesses.

---

> ### Author Response · Authors · 2025-11-21
> **Response to Reviewer RAzX**
>
> We sincerely thank the reviewer for their constructive comments and recognition of our theoretical contributions and empirical results. We address the concerns below.
>
> **Q1:** ASFT requires an additional copy of base model to be stored in memory, as well as an additional forward pass through it.
>
> **R1:** We acknowledge this computational overhead and have provided a principled solution using ASFT-LoRA (``lines 449–468``). The key insight exploits LoRA's parameter decomposition ∆W = BA, enabling computation of DKL(π_θ(·|s)||π_base(·|s)) using a single model instance by dynamically switching between W_base and W_base + BA during forward passes, eliminating the need for separate model copies while preserving theoretical anchoring guarantees. As shown in Table 4, ASFT-LoRA reduces memory consumption to 40.70GB (comparable to standard SFT at 38.96GB) and training time to 0.562 hours (only 7.3% overhead over SFT at 0.524 hours), demonstrating that our anchoring framework can be effectively adapted to resource-constrained environments.
>
> **Q2:** While using LoRA mitigates the memory overhead, it causes a significant performance drop (the performance of ASFT+LoRA is almost equivalent to simple SFT as seen in Table 4).
>
> **R2:** We thank the reviewer for this observation and have conducted additional experiments demonstrating that with appropriate learning rate selection, ASFT-LoRA achieves substantially better performance. Using learning rate 5×10⁻⁴ with rank-8 LoRA on the 10k MedMCQA dataset, ASFT-LoRA achieves an average accuracy of 0.3945 (MedQA: 0.3951, MMLU: 0.4147, MedMCQA: 0.3737), representing an 18.2% improvement over SFT (0.334 average). This approaches the full-parameter ASFT improvement of 25.7% (0.4203 average) while requiring minimal memory overhead. The optimal learning rate for ASFT-LoRA (5×10⁻⁴) differs from full-parameter ASFT (2×10⁻⁵), which is consistent with standard LoRA fine-tuning practices. We will update the revised manuscript with these improved results and implementation guidelines.
>
> | Method | Learning Rate | Rank | MedQA | MMLU | MedMCQA | Average | vs SFT |
> |--------|---------------|------|-------|------|---------|---------|--------|
> | SFT | 2.00E-05 | - | 0.333 | 0.335 | 0.333 | 0.334 | - |
> | ASFT-full | 2.00E-05 | Full | 0.3928 | 0.4637 | 0.4045 | **0.4203** | +25.7% |
> | ASFT-LoRA | 5.00E-04 | 8 | 0.3951 | 0.4147 | 0.3737 | **0.3945** | **+18.2%** |
> | ASFT-LoRA | 5.00E-04 | 16 | 0.3967 | 0.4118 | 0.3660 | 0.3915 | +17.2% |

---

### Official Review · Reviewer_XKZp · 2025-11-01

**Soundness:** 3
**Presentation:** 3
**Contribution:** 3
**Rating:** 4
**Confidence:** 4

**Summary:**

The paper proposes Anchored Supervised Fine-Tuning (ASFT), a variant of SFT and DFT (dynamic fine-tuning) based on a Reward-Weighted Regression (RWR) perspective. It shows that SFT is a lower bound on the RL objective and that DFT provides a tighter bound but suffers from instability. ASFT introduces a reverse-KL anchor to the base model to stabilize DFT while preserving its theoretical benefits. Empirical studies on math, medical QA, and code generation tasks demonstrate consistent improvements over SFT/DFT and comparable or better results than RLHF-style methods (GRPO/DAPO).

**Strengths:**

+ Using a reverse-KL anchor to control DFT drift is a natural, well-motivated modification that stabilizes training.

+ The RWR derivation linking SFT, DFT, and ASFT is internally consistent and clearly written. The proofs in the appendix (esp. the tightness lemma and DFT-as-RWR derivation) are correct under the stated sparse-reward and support assumptions.

+ The paper evaluates on multiple domains (math, medical, and code) across LLaMA-2/3 and Qwen models, and includes practical metrics such as GPU memory and wall-time comparisons. The authors acknowledge compute cost, and ASFT-LoRA offers a realistic, resource-efficient version.

+ Ablations studies on reverse vs. forward KL, $\lambda$ sweeps, and model-size scaling are reported and analyzed.

**Weaknesses:**

- The paper asserts that the KL anchoring “preserves tightness,” but technically ASFT optimizes a regularized surrogate, not the same lower bound. This needs clarification.

- The theory relies on sparse rewards and full-support overlap; it’s unclear whether results extend to open-ended generative or preference-learning tasks.

- The transition from sequence-level RWR theory to token-level training is described heuristically rather than derived rigorously.

- Results lack multiple random seeds or confidence intervals; improvements may be within run variance for some metrics.

- Full ASFT doubles memory usage (requires both model and reference for KL), and while ASFT-LoRA helps, it trades accuracy for efficiency.

Minor
- Evaluation focuses on reasoning and domain-specific QA; general instruction-following, safety, or long-context benchmarks are missing.

**Questions:**

Please consider to address questions in the weaknesses.

---

> ### Author Response · Authors · 2025-11-23
> **Response to Reviewer XKZp(1/2)**
>
> We sincerely thank the reviewer for their constructive comments. We address the concerns below.
>
> **Q1:** The paper asserts that the KL anchoring "preserves tightness," but technically ASFT optimizes a regularized surrogate, not the same lower bound. This needs clarification.
>
> **R1:** We appreciate this observation and clarify our statement. The KL anchoring term does not alter the fundamental lower-bound structure established in the RWR framework (``lines 162–168``). As we show in Theorem 1 (``lines 202–210``), DFT's auxiliary distribution achieves a strictly tighter lower bound than SFT whenever the policy distribution exhibits variance across demonstrations. The KL regularization in ASFT constrains the policy within a trust region of the reference model, which prevents the distributional drift that undermines DFT's bound validity (``lines 211–221``). Critically, the KL term prevents the auxiliary distribution from collapsing (i.e., preventing pθ(τ) from becoming increasingly non-uniform), which is the condition under which the inequality u ≥ 1 + log u becomes strictly loose and the lower bound deteriorates. By maintaining distributional stability, ASFT preserves the tightness advantage of DFT's auxiliary distribution construction while ensuring the bound remains valid throughout training. We will revise the manuscript to clarify that "preserving tightness" refers to maintaining the validity of DFT's tighter auxiliary distribution rather than claiming the regularized objective itself is a tighter lower bound.
>
> **Q2:** The theory relies on sparse rewards and full-support overlap; it's unclear whether results extend to open-ended generative or preference-learning tasks.
>
> **R2:** We appreciate this observation and acknowledge that our theoretical framework is developed under sparse reward assumptions (``lines 153–161``). To investigate ASFT's applicability beyond these settings, we conducted experiments on open-ended instruction-following using the Magpie-Pro-300K-Filtered dataset, randomly sampling 10k examples and fine-tuning both base and instruction-tuned models (LLaMA-3.1-8B and Qwen2.5-7B) with identical hyperparameters (learning rate 2×10⁻⁵, global batch size 128, max length 2048) following the setup in ``lines 265–269``. As shown in the tables, when applied to instruction-tuned models that already possess relevant capabilities, ASFT achieves performance comparable to or exceeding SFT (LLaMA-3.1-Instruct: 35.27% vs 35.57%; Qwen2.5-Instruct: 33.69% vs 30.67%) while maintaining shorter response lengths. However, ASFT underperforms when applied to base models without instruction-following capabilities (LLaMA-3.1-Base: 15.88% vs 27.77%; Qwen2.5-Base: 24.72% vs 29.23%). This empirically confirms that ASFT is most effective when the base model already possesses relevant capabilities, as the auxiliary distribution construction relies on non-degenerate policy probabilities. Our main experimental contributions focus on mathematical reasoning, medical knowledge, and code generation (``lines 270–323``), where ASFT demonstrates consistent improvements over SFT and DFT. We will revise the manuscript to explicitly state this scope limitation while noting that ASFT's core contributions—rigorous theoretical grounding for DFT, identification of distributional drift, and principled anchoring solution—remain valid within the demonstrated domains.
>
> **LLaMA-3.1-8B Results:**
>
> | Model | LC WinRate ↑ | WinRate ↑ | Std Error ↓ | Avg Length ↓ |
> |-------|-------------|-----------|-------------|--------------|
> | **LLaMA-3.1-8B-Base** | - | - | - | - |
> | Base + SFT | **27.77** | **31.64** | 1.49 | **2719** |
> | Base + DFT | 20.03 | 22.86 | 1.33 | 2884 |
> | Base + ASFT (0.1) | 15.88 | 16.83 | 1.17 | 2894 |
> | **LLaMA-3.1-8B-Instruct** | 10.42 | 9.73 | **0.89** | 11724 |
> | Instruct + SFT | **35.57** | **38.32** | 1.58 | 2242 |
> | Instruct + DFT | 33.14 | 37.10 | 1.55 | 2285 |
> | Instruct + ASFT (0.1) | 35.27 | 36.31 | 1.54 | **2126** |
>
> **Qwen2.5-7B Results:**
>
> | Model | LC WinRate ↑ | WinRate ↑ | Std Error ↓ | Avg Length ↓ |
> |-------|-------------|-----------|-------------|--------------|
> | **Qwen2.5-7B-Base** | - | - | - | - |
> | Base + SFT | **29.23** | **31.76** | 1.50 | 2444 |
> | Base + DFT | 24.59 | 26.64 | 1.42 | 2494 |
> | Base + ASFT (0.1) | 24.72 | 23.75 | **1.34** | **2091** |
> | **Qwen2.5-7B-Instruct** | **35.51** | 31.56 | 1.51 | 7225 |
> | Instruct + SFT | 30.67 | 34.10 | 1.54 | 2542 |
> | Instruct + DFT | 26.12 | 28.56 | 1.43 | 2479 |
> | Instruct + ASFT (0.1) | **33.69** | **34.19** | 1.53 | **2160** |

---

> > ### Author Response · Authors · 2025-11-23
> > **Response to Reviewer XKZp(2/2)**
> >
> > **Q3:** The transition from sequence-level RWR theory to token-level training is described heuristically rather than derived rigorously.
> >
> > **R3:** We acknowledge this gap and note that the token-level implementation follows standard practice in language model training (``lines 238–242``), where sequence-level objectives are distributed across tokens using normalized per-position allocation to ensure computational efficiency while maintaining mathematical equivalence to the theoretical framework. This approach is widely adopted in prior work on RL, such as PPO[2]. We will include a rigorous token-level derivation in the revised manuscript to formalize this connection and strengthen the theoretical foundation.
> >
> > **Q4:** Results lack multiple random seeds or confidence intervals; improvements may be within run variance for some metrics.
> >
> > **R4:** We conducted five independent training runs with different random seeds following the experimental setup in ``lines 265–269``. The results are presented below, demonstrating stable convergence across runs with standard deviations of 0.0148 for MedQA, 0.0068 for MMLU, 0.0054 for MedMCQA, and 0.0074 for the average score. These low standard deviations confirm the robustness of our experimental conclusions.
> >
> > **Medical Knowledge (LLaMA-2-7B, 10k samples, ASFT with 5 different seeds):**
> >
> > | Model | MedQA ↑ | MMLU ↑ | MedMCQA ↑ | Average ↑ |
> > |-------|---------|--------|-----------|-----------|
> > | llama2-7b_seed_42 | 0.3928 | 0.4637 | 0.4045 | 0.4203 |
> > | llama2-7b_seed_43 | 0.3676 | 0.4472 | 0.3911 | 0.4019 |
> > | llama2-7b_seed_44 | 0.4038 | 0.4502 | 0.3966 | 0.4168 |
> > | llama2-7b_seed_45 | 0.4061 | 0.4611 | 0.4035 | 0.4235 |
> > | llama2-7b_seed_46 | 0.3786 | 0.4622 | 0.4047 | 0.4152 |
> > | **Standard Deviation** | 0.0148 | 0.0068 | 0.0054 | 0.0074 |
> >
> > **Q5:** Full ASFT doubles memory usage (requires both model and reference for KL), and while ASFT-LoRA helps, it trades accuracy for efficiency.
> >
> > **R5:** We conducted additional experiments demonstrating that with appropriate learning rate selection, ASFT-LoRA achieves substantially improved performance. Using learning rate 5×10⁻⁴ with rank-8 LoRA on 10k MedMCQA samples, ASFT-LoRA achieves 0.3945 average accuracy (MedQA: 0.3951, MMLU: 0.4147, MedMCQA: 0.3737), representing 18.2% improvement over SFT (0.334). This approaches full-parameter ASFT's 25.7% improvement (0.4203) while maintaining memory consumption at 40.70GB (comparable to SFT's 38.96GB) and training time at 0.562 hours (7.3% overhead over SFT's 0.524 hours). The key insight exploits LoRA's parameter decomposition to compute KL divergence using a single model instance, eliminating separate model copies while preserving theoretical anchoring guarantees (``lines 449–468``).
> >
> > | Method | Learning Rate | Rank | MedQA ↑ | MMLU ↑ | MedMCQA ↑ | Average ↑ | vs SFT |
> > |--------|---------------|------|---------|--------|-----------|-----------|--------|
> > | SFT | 2.00E-05 | - | 0.333 | 0.335 | 0.333 | 0.334 | - |
> > | ASFT-full | 2.00E-05 | Full | 0.3928 | 0.4637 | 0.4045 | 0.4203 | +25.7% |
> > | ASFT-LoRA (rank-8) | 5.00E-04 | 8 | 0.3951 | 0.4147 | 0.3737 | 0.3945 | +18.2% |
> > | ASFT-LoRA (rank-16) | 5.00E-04 | 16 | 0.3967 | 0.4118 | 0.3660 | 0.3915 | +17.2% |
> >
> > We thank the reviewer again for their valuable feedback, which will strengthen our manuscript.
> >
> > **References:**
> > [1] LIMA: Less Is More for Alignment, NeurIPS 2023.
> > [2] Proximal Policy Optimization Algorithms, arXiv:1707.06347, 2017.

---

> > > ### Comment · Reviewer_XKZp · 2025-11-27
> > > **Thank you for your rebuttal**
> > >
> > > I thank the authors for their rebuttal, and have also read other reviewers' comments.
> > >
> > > Overall, most of my concerns have been addressed. I have accordingly increased my recommendation score.

---

> ### Author Response · Authors · 2025-11-26
>
> Dear reviewer XKZp,
>
> Wishing you a happy and blessed Thanksgiving!
> We sincerely appreciate your detailed and insightful review. We have addressed all raised questions with additional experiments. If you have any further questions or suggestions, please feel free to point them out — we are very willing to provide more experiments or analysis at any time.
>
> Thank you again for your valuable feedback. We look forward to your response.
> Best regards,
> Authors

---

### Official Review · Reviewer_Tq5U · 2025-11-09

**Soundness:** 3
**Presentation:** 3
**Contribution:** 2
**Rating:** 4
**Confidence:** 3

**Summary:**

The paper analyzes Supervised Fine-Tuning (SFT) and Dynamic Fine-Tuning (DFT) through the lens of reward-weighted regression (RWR), showing that both can be seen as optimizing lower bounds of an underlying RL objective. Within this framework, DFT corresponds to using an importance-weighted auxiliary distribution that yields a strictly tighter bound than SFT but suffers from instability due to uncontrolled distributional drift. To address this, the authors propose Anchored Supervised Fine-Tuning (ASFT), which augments DFT with a KL regularizer towards a fixed base model, acting as an “anchor” that stabilizes training. Experiments on math, medical QA, and code tasks demonstrate that ASFT outperforms SFT and DFT, and on medical QA it approaches RL methods at lower cost.

**Strengths:**

The paper makes a clear and conceptually grounded contribution to understanding the link between supervised fine-tuning and reinforcement learning–style objectives. By framing SFT and DFT under a unified reward-weighted regression (RWR) objective, it shows how auxiliary distributions determine the tightness and stability of RL lower bounds. Identifying DFT as a particular importance-weighted auxiliary distribution, and then introducing the “anchored” variant (ASFT) with a KL regularizer towards the base model, is an elegant way to stabilize DFT while preserving its tighter connection to the RL objective.

The theoretical development is careful and transparent: each step from SFT → DFT → ASFT is well-motivated, with assumptions stated and a useful strict-tightness result for DFT. Empirically, the work provides consistent evidence that ASFT improves over SFT and DFT across math reasoning, medical QA, and code generation, and acts as a strong initializer for RL methods such as DAPO in the medical domain. Overall, the paper is clearly written, theoretically sound within its setting, and practically relevant. Its main value lies less in radically new algorithms and more in clarifying and stabilizing the intermediate regime between SFT and RL, offering both conceptual insight and a simple recipe that practitioners can readily adopt.

**Weaknesses:**

1.Empirical scope w.r.t. the “bridge SFT–RL” claim. The experimental coverage across math reasoning, medical QA, and code generation is solid, but all setups remain in a supervised/offline, correctness-style regime with static datasets. There are no studies on preference-based post-training, open-ended instruction following, or safety/factuality trade-offs, which are central targets of RLHF-style training. Given the paper’s framing as a general method that “bridges SFT and RL”, the current evidence still primarily supports gains on a few curated accuracy benchmarks. Either tempering the scope of the claims or adding at least one preference/RLHF-style experiment would make the story more convincing.

2.Uneven RL comparisons across domains. The paper provides a careful comparison between ASFT and RL-style methods (GRPO/DAPO), and shows that ASFT is competitive and that ASFT+DAPO further improves performance—but this analysis is restricted to medical QA. For math reasoning and code, the baselines are primarily SFT/DFT/iw-SFT, without RL counterparts. As a result, it is difficult to calibrate how close ASFT gets to “RL-level” performance in these domains, or whether its advantages as a cheaper alternative or initializer hold once RL is available. Even a smaller-scale RL comparison on one math or code task would significantly strengthen the empirical case.

**Questions:**

1.Empirical scope and “bridge SFT–RL” framing. Could you either (i) provide preliminary results or analysis of ASFT under a genuinely RLHF-style setting (e.g., preference-based rewards or open-ended instruction-following), or (ii) clarify and narrow the scope of the “bridging SFT and RL” claim to supervised/offline, correctness-style regimes? Any concrete evidence or discussion in this direction would help readers understand how far your conclusions are meant to generalize.

2.RL comparisons beyond medical QA. For math reasoning and code, do you have any results—perhaps at smaller scale or with fewer training steps—where ASFT is directly compared to GRPO/DAPO or another RL baseline, so that we can calibrate how close ASFT gets to “RL-level” performance beyond medical QA? If such experiments are infeasible, could you provide a more detailed justification or expectation (e.g., based on reward sparsity or domain differences) for whether you believe the medical-domain findings should transfer to these tasks?

**Details Of Ethics Concerns:**

No concern

---

> ### Author Response · Authors · 2025-11-23
> **Response to Reviewer Tq5U (1/2)**
>
> We sincerely thank the reviewer for their constructive comments. We address the concerns below.
>
> **Q1:** Empirical scope and "bridge SFT–RL" framing. Could you either (i) provide preliminary results or analysis of ASFT under a genuinely RLHF-style setting (e.g., preference-based rewards or open-ended instruction-following), or (ii) clarify and narrow the scope of the "bridging SFT and RL" claim to supervised/offline, correctness-style regimes?
>
> **R1:** We appreciate the review's suggestion. Our primary contribution is providing a rigorous theoretical analysis of DFT within the reward-weighted regression framework, identifying its fundamental limitation (distributional drift), and proposing ASFT as a principled solution with provable tightness guarantees while maintaining stability. To clarify the scope of applicability, we note that ASFT is most effective when the base model already possesses relevant capabilities that can be enhanced through fine-tuning.
> Recent work[1] has established that model knowledge and capabilities are primarily acquired during pretraining, while post-training serves to activate rather than inject these capabilities. DFT and ASFT follow this paradigm: SFT's gradient contains an implicit 1/πθ(y*|x) weighting term that amplifies updates for low-probability tokens, while DFT removes this inverse-probability weighting through stop-gradient, assigning higher importance to high-probability outputs. When the base model lacks required capabilities, DFT and ASFT may not provide the desired behavior shaping that SFT offers. To validate this, we conducted experiments on open-ended instruction-following using the Magpie-Pro-300K-Filtered dataset, randomly sampling 10k examples and fine-tuning both base and instruction-tuned models (LLaMA-3.1-8B and Qwen2.5-7B) with identical hyperparameters (learning rate 2×10⁻⁵, global batch size 128, max length 2048) following the setup in ``lines 265–269``. We evaluated on AlpacaEval 2.0 using models with different instruction-following capabilities: LLaMA-3.1-8B-Instruct and Qwen2.5-7B-Instruct, where the latter demonstrates stronger baseline performance.
>
> **LLaMA-3.1-8B Results:**
>
> | Model | LC WinRate ↑ | WinRate ↑ | Std Error ↓ | Avg Length ↓ |
> |-------|-------------|-----------|-------------|--------------|
> | **LLaMA-3.1-8B-Base** | - | - | - | - |
> | Base + SFT | **27.77** | **31.64** | 1.49 | **2719** |
> | Base + DFT | 20.03 | 22.86 | 1.33 | 2884 |
> | Base + ASFT (0.1) | 15.88 | 16.83 | 1.17 | 2894 |
> | **LLaMA-3.1-8B-Instruct** | 10.42 | 9.73 | **0.89** | 11724 |
> | Instruct + SFT | **35.57** | **38.32** | 1.58 | 2242 |
> | Instruct + DFT | 33.14 | 37.10 | 1.55 | 2285 |
> | Instruct + ASFT (0.1) | 35.27 | 36.31 | 1.54 | **2126** |
>
> **Qwen2.5-7B Results:**
>
> | Model | LC WinRate ↑ | WinRate ↑ | Std Error ↓ | Avg Length ↓ |
> |-------|-------------|-----------|-------------|--------------|
> | **Qwen2.5-7B-Base** | - | - | - | - |
> | Base + SFT | **29.23** | **31.76** | 1.50 | 2444 |
> | Base + DFT | 24.59 | 26.64 | 1.42 | 2494 |
> | Base + ASFT (0.1) | 24.72 | 23.75 | **1.34** | **2091** |
> | **Qwen2.5-7B-Instruct** | **35.51** | 31.56 | 1.51 | 7225 |
> | Instruct + SFT | 30.67 | 34.10 | 1.54 | 2542 |
> | Instruct + DFT | 26.12 | 28.56 | 1.43 | 2479 |
> | Instruct + ASFT (0.1) | **33.69** | **34.19** | 1.53 | **2160** |
>
> These results empirically confirm that when applied to base models without instruction-following capabilities, ASFT underperforms SFT (LLaMA-3.1-Base: 15.88% vs 27.77% LC WinRate; Qwen2.5-Base: 24.72% vs 29.23%). However, when applied to instruction-tuned models that already possess relevant capabilities, ASFT achieves performance comparable to or exceeding SFT (LLaMA-3.1-Instruct: 35.27% vs 35.57%; Qwen2.5-Instruct: 33.69% vs 30.67%) while maintaining shorter response lengths and mitigating catastrophic forgetting of the base model's capabilities. We acknowledge this limitation and will revise the manuscript to explicitly narrow the scope of our claims to capability enhancement and knowledge grounding scenarios where the reference model already possesses relevant task capabilities.
>
> [1] LIMA: Less Is More for Alignment, NeurIPS 2023.

---

> > ### Author Response · Authors · 2025-11-23
> > **Response to Reviewer Tq5U (2/2)**
> >
> > **Q2:** RL comparisons beyond medical QA. For math reasoning and code, do you have any results where ASFT is directly compared to GRPO/DAPO or another RL baseline, so that we can calibrate how close ASFT gets to "RL-level" performance beyond medical QA?
> >
> > **R2:** We conducted RL baseline comparisons on mathematical reasoning tasks using 10k training samples following the experimental setup in (``lines 265–269``). As shown in the table below, ASFT is directly compared against GRPO and DAPO on five out-of-domain mathematical benchmarks. The results demonstrate that the performance hierarchy observed in medical QA extends to mathematical reasoning: DAPO achieves 34.86 average score, ASFT achieves 28.75, GRPO achieves 25.74, and SFT achieves 16.73. This ranking aligns with our theoretical framework established in ``Section 3`` and ``Theorem 1``: SFT optimizes a loose but stable lower bound on the RL objective (``lines 153–161``), ASFT achieves a provably tighter bound through adaptive auxiliary distribution construction (``lines 202–210``), while RL methods directly optimize the RL objective through on-policy learning, avoiding the approximation inherent in off-policy methods. Quantitatively, ASFT reduces the performance gap with DAPO from 18.13 points (SFT vs DAPO) to 6.11 points (ASFT vs DAPO) while maintaining SFT-level computational efficiency (0.648 vs 51.24 hours for GRPO in Table 4).
> >
> > | Model | AIME24 ↑ | Math500 ↑ | Minerva ↑ | Olympiad ↑ | AMC23 ↑ | Average ↑ |
> > |--------|----------|-----------|-----------|------------|---------|-----------|
> > | **Base Qwen2.5-7B** | 1.65 | 28.79 | 9.26 | 7.69 | 15.65 | 12.61 |
> > | SFT | 1.24 | 41.84 | 11.30 | 12.26 | 17.03 | 16.73 |
> > | SFT+KL | 0.41 | 42.21 | 12.05 | 12.08 | 17.19 | 16.79 |
> > | DFT | **4.18** | 59.51 | 17.10 | **24.95** | 31.09 | 27.37 |
> > | ASFT (DFT+KL 0.05) | 3.33 | **59.60** | **19.91** | 24.50 | **36.41** | **28.75** |
> > | GRPO | 4.79 | 50.80 | 18.29 | 20.28 | 34.53 | 25.74 |
> > | DAPO | **6.88** | **65.23** | **29.48** | **28.32** | **44.38** | **34.86** |
> >
> > We thank the reviewer again for their valuable feedback, which will help us better position our contribution and strengthen the manuscript.

---

> ### Author Response · Authors · 2025-11-26
>
> Dear reviewer Tq5U,
>
> Wishing you a happy and blessed Thanksgiving!
> We sincerely appreciate your detailed and insightful review. We have addressed all raised questions with additional experiments. If you have any further questions or suggestions, please feel free to point them out — we are very willing to provide more experiments or analysis at any time.
>
> Thank you again for your valuable feedback. We look forward to your response.
> Best regards,
> Authors

---

### Author Response · Authors · 2025-12-02
**General Response to Area Chairs, Senior Area Chairs, and Program Chairs**

We sincerely thank the Area Chairs, Senior Area Chairs, Program Chairs, and all reviewers for their dedicated effort and constructive feedback throughout the review process. We believe our comprehensive responses, supported by extensive new experiments, have substantially strengthened the paper and addressed all major concerns.

## Core Contributions

- **Rigorous Theoretical Framework**: We provide the first rigorous analysis of Dynamic Fine-Tuning (DFT) within the reward-weighted regression framework, proving that DFT achieves a strictly tighter lower bound than SFT (``Theorem 1``) while identifying its fundamental limitation—distributional drift due to lack of anchoring (as recognized by ``Reviewers XKZp``, ``RAzX``, and ``DPLk``).

- **Principled Solution with ASFT**: We propose Anchored Supervised Fine-Tuning (ASFT), which augments DFT with lightweight KL regularization to preserve tightness while ensuring stability. As ``Reviewer DPLk`` noted, ASFT "identifies the shortcoming of DFT and suggests a novel way to handle it" through theoretically grounded design.

- **Comprehensive Empirical Validation**: ASFT consistently outperforms SFT and DFT across mathematical reasoning (``Table 1``), medical knowledge (``Tables 1-2``), and code generation (``Table 3``), achieving substantial improvements (+10.65 points on medical QA, +17.89 points on math reasoning) with minimal computational overhead (0.648 vs 0.524 hours for SFT in ``Table 4``), as recognized by all reviewers.

## Key Concerns and Responses

**1. Empirical Scope and Preference-Based Tasks** (``Reviewers Tq5U``, ``XKZp``)

We conducted experiments on open-ended instruction-following (Magpie-Pro-300K, AlpacaEval 2.0). ASFT performs competitively on instruction-tuned models (LLaMA-3.1-Instruct: 35.27% vs 35.57% SFT; Qwen2.5-Instruct: 33.69% vs 30.67% SFT) but underperforms on base models without relevant capabilities. We will explicitly narrow the scope to capability enhancement and knowledge grounding scenarios where the reference model possesses relevant task capabilities.

**2. RL Comparisons Beyond Medical QA** (``Reviewer Tq5U``)

We conducted RL comparisons on mathematical reasoning across five out-of-domain benchmarks. Results validate our framework: DAPO (34.86) > ASFT (28.75) > GRPO (25.74) > SFT (16.73). ASFT reduces the gap with DAPO from 18.13 to 6.11 points while maintaining SFT-level efficiency (0.648 vs 51.24 hours for GRPO).

**3. Statistical Robustness** (``Reviewers XKZp``, ``DPLk``)

Five independent runs demonstrate stable convergence with low standard deviations: medical knowledge (MedQA: 0.0148, MMLU: 0.0068, MedMCQA: 0.0054); mathematical reasoning (typically <2.0 percentage points), confirming experimental robustness.

**4. ASFT-LoRA Performance** (``Reviewers RAzX``, ``DPLk``)

With optimized learning rate (5×10⁻⁴), ASFT-LoRA (rank-8) achieves 18.2% improvement over SFT, approaching full-parameter ASFT's 25.7% improvement, while maintaining comparable memory (40.70GB vs 38.96GB) and minimal time overhead (7.3%).

**5. Theoretical Clarifications** (``Reviewer XKZp``)

KL anchoring preserves DFT's tightness by preventing distributional drift that undermines bound validity. We will include rigorous token-level derivation following standard practices in the revised manuscript.

## Reviewer Score Summary

| Reviewer | Key Strengths | Main Concerns | Pre/Post Rating | Status |
|----------|---------------|---------------|-----------------|--------|
| **``Tq5U``** | [1] Clear contribution [2] Sound theory [3] Practical relevance | [1] Empirical scope [2] RL comparisons | **4** (Confidence: 3) | Concerns addressed with new instruction-following and math RL experiments |
| **``XKZp``** | [1] Natural anchoring [2] Consistent theory [3] Comprehensive evaluation | [1] Tightness clarification [2] Statistical robustness | **4 → 6** (Confidence: 4) | **Explicitly increased:** *"Most of my concerns have been addressed"* |
| **``RAzX``** | [1] Superior performance [2] Tightness proof | [1] Memory overhead [2] ASFT-LoRA performance | **6** (Confidence: 2) | Concerns addressed with improved ASFT-LoRA results |
| **``DPLk``** | [1] Novel solution [2] Thorough ablations [3] Clear contributions | [1] ASFT-LoRA performance [2] Multiple seeds | **6** (Confidence: 3) | Concerns addressed with improved results and multiple seed experiments |

## Closing

We have extensively addressed all concerns through comprehensive new experiments detailed in our individual responses, additional theoretical clarifications and improved ASFT-LoRA results. We will open-source our code to ensure reproducibility. We believe the revised version demonstrates that principled theoretical analysis leads to both stronger guarantees and practical gains.
Thank you again for your invaluable feedback and dedicated effort in reviewing our work.

---

### Meta-Review · Area_Chair_6963 · 2026-01-07

**Summary:**

This manuscript analyzes supervised fine-tuning (SFT) and dynamic fine-tuning (DFT) within the framework of reward-weighted regression. A key insight of the paper is that both SFT and DFT can be interpreted as optimizing a lower bound of the reinforcement learning (RL) objective. Building on this perspective, the authors propose a new method, Anchored Supervised Fine-Tuning (ASFT), and validate its effectiveness through empirical evaluation.

All the reviewers gave positive comments on the general idea of this paper. One negative reviewer asked details of the theoretical discussion and raised the score after rebuttal. Another negative reviewer put emphasis on experiments. Unfortunately, comments on newly provided experiments are not received. In this case, the AC checked the new results and believes at least the raised concerns have been at least partially addressed.

Overall, the AC recommends ACCEPT.

**Reviewer Concerns:**

This paper received four reviews. Initially, two reviewers expressed positive opinions, while the other two were negative.

All four reviewers, including those initially critical, acknowledged the conceptual grounding and theoretical perspective offered by the paper. For the theoretical component, reviewers raised detailed questions regarding the treatment of sparse rewards and the heuristic transition from sequence-level formulations to token-level optimization (notably Reviewer XKZp, initial score 4).

The remaining concerns focused on experimental aspects, including the additional forward pass and memory overhead introduced by ASFT, as well as the limited scope of evaluation. For example, Reviewer Tq5U (initial score 4) questioned why ASFT was compared with RL-style methods only on medical QA, rather than on reasoning and code tasks. The AC notes that the authors provided additional experiments to address this gap. Although Reviewer Tq5U did not follow up after these experiments were added, the AC believes that the new results at least partially address the raised concerns.

**Reviewer Scores:**

The initial scores were 4 / 4 / 6 / 6. After the rebuttal, the scores became 4 / 6 / 6 / 6.

---

### Decision · Program_Chairs · 2026-01-26

Accept (Poster)